# FreqPrior: Improving Video Diffusion Models with Frequency Filtering Gaussian Noise

**Yunlong Yuan[1], Yuanfan Guo[2], Chunwei Wang[2], Wei Zhang[2], Hang Xu[2], Li Zhang[1]***

[1]School of Data Science, Fudan University    [2]Noah's Ark Lab, Huawei

https://github.com/fudan-zvg/FreqPrior

## ABSTRACT

Text-driven video generation has advanced significantly due to developments in diffusion models. Beyond the training and sampling phases, recent studies have investigated noise priors of diffusion models, as improved noise priors yield better generation results. One recent approach employs the Fourier transform to manipulate noise, marking the initial exploration of frequency operations in this context. However, it often generates videos that lack motion dynamics and imaging details. In this work, we provide a comprehensive theoretical analysis of the variance decay issue present in existing methods, contributing to the loss of details and motion dynamics. Recognizing the critical impact of noise distribution on generation quality, we introduce FreqPrior, a novel noise initialization strategy that refines noise in the frequency domain. Our method features a novel filtering technique designed to address different frequency signals while maintaining the noise prior distribution that closely approximates a standard Gaussian distribution. Additionally, we propose a partial sampling process by perturbing the latent at an intermediate timestep while finding the noise prior, significantly reducing inference time without compromising quality. Extensive experiments on VBench demonstrate that our method achieves the highest scores in both quality and semantic assessments, resulting in the best overall total score. These results highlight the superiority of our proposed noise prior.

## 1 INTRODUCTION

Benefiting from notable advancements of diffusion models (Sohl-Dickstein et al., 2015; Ho et al., 2020; Song et al., 2021b) alongside the expansion of large video datasets (Bain et al., 2021; Schuhmann et al., 2022), text-to-video generation has experienced remarkable progress (Ho et al., 2022a; Wu et al., 2022a; Blattmann et al., 2023; Ge et al., 2023; Guo et al., 2024; Singer et al., 2023; Wang et al., 2023; Chen et al., 2023). In ordinary videos, the content between successive frames often shows high similarity, allowing the video to be considered as a sequence of images with motion information. Leveraging this characteristic, the architecture of video diffusion models (Blattmann et al., 2023; Wang et al., 2023; Hong et al., 2023; Guo et al., 2024) commonly incorporates temporal or motion layers into existing image diffusion models. In addition to model architecture, some studies, inspired by the consistent patterns observed across video frames, investigate the relationships within the initial noise prior. Consequently, alongside research focusing on the training and sampling phases (Song et al., 2021a; Karras et al., 2022; Lu et al., 2022; Salimans & Ho, 2022; Song et al., 2023), another important line of research in video diffusion models is to explore noise initialization strategies, since improved noise prior can potentially yield better generation results.

Several efforts have been made to explore the noise prior, as the initial noise significantly impacts the generated outcomes (Ge et al., 2023; Qiu et al., 2024; Chang et al., 2024; Gu et al., 2023; Mao et al., 2024; Wu et al., 2024). PYoCo (Ge et al., 2023) discovers that the noise maps corresponding to different frames, derived from a pre-trained image diffusion model, cluster in t-SNE space (Van der Maaten & Hinton, 2008), indicating a strong correlation along the temporal dimension. Based on this observation, it introduces two kinds of noise prior with correlations on the frame dimension.

---

*Corresponding author lizhangfd@fudan.edu.cn.

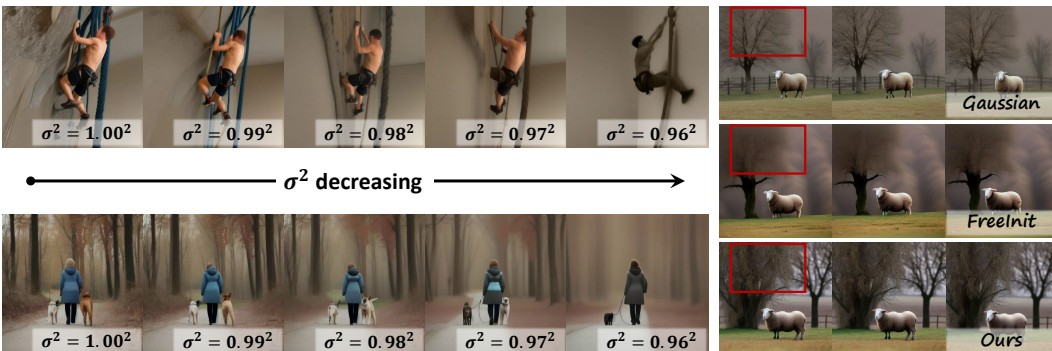

Figure 1: *(Left)* **Generated video frames corresponding to Gaussian noise with different variances.** As the variance, denoted as $\sigma^2$, decreases from $1.00^2$ to $0.96^2$, the imaging quality deteriorates and background details are gradually lost. *(Right)* **Comparisons of our method against the FreeInit and standard Gaussian noise.** The frames generated using FreeInit appear overly smooth and blurred in the area of the highlighted red box, whereas our method preserves rich image details.

However, this change in the noise prior requires massive fine-tuning. FreeInit (Wu et al., 2024) investigates the low-frequency signal leakage phenomenon in the noise, as also demonstrated in the image domain (Lin et al., 2024), and finds that the denoising process is significantly influenced by the low-frequency components of initial noise. Leveraging these insights, it uses frequency filtering on the noise prior to enhance the temporal consistency of generated videos. However, despite its efforts, the generated videos suffer from excessive smoothness, limited motion dynamics, and a lack of details. Moreover, additional iterations are necessary to refine the noise, with a full sampling process conducted in each iteration, making FreeInit (Wu et al., 2024) quite time-consuming.

To address this gap, we conduct a mathematical analysis and provide theoretical justification. Our analysis identifies the variance decay issue existing in FreeInit (Wu et al., 2024). As depicted in Figure 1, we investigate the significance of the distribution of the initial noise for diffusion models. The impact of the variance on the quality of generated videos is evident. As $\sigma$ decreases from 1 to 0.96, there is a progressive loss of details alongside a reduction in motion dynamics. The frames generated by FreeInit (Wu et al., 2024) are overly smooth and lack details, as the refined noise deviates from the standard Gaussian distribution, resulting in variance decay. Therefore, it is critically important for diffusion models that the noise prior follows standard Gaussian distribution.

In this work, we introduce a novel noise prior called **FreqPrior**. At the core of our approach is the noise refinement stage, where we propose a novel frequency filtering method designed for noise, which essentially is random variables. During this stage, we retain the low-frequency signals while enriching high-frequency signals in the frequency domain, thereby reducing the covariance error and ensuring that the distribution of our refined noise approximates a standard Gaussian distribution. As illustrated in Figure 1, our method does not suffer from the detail loss issue present in FreeInit (Wu et al., 2024). Additionally, retaining low-frequency signals enhances semantic fidelity. Furthermore, to obtain the noise prior, we adjust the diffusion process by perturbing the latent at an intermediate step, resulting in significant time savings without compromising the quality of the generation results. We conduct extensive experiments on Vbench (Huang et al., 2024b), a comprehensive benchmark, to assess the quality of generated videos. The results demonstrate that our method effectively addresses the issue of limited dynamics while improving the overall quality. Moreover, our approach outperforms the best on VBench, highlighting the superiority of our method. Additionally, our method achieves a time-saving of nearly 23% compared to FreeInit (Wu et al., 2024).

In summary, our contributions are as follows: **(i):** We propose a novel frequency filtering method designed to refine the noise, acquiring a better prior, termed **FreqPrior**. We provide a rigorous theoretical analysis of the distribution of our prior. Numerical experiments reveal the covariance error of our method is negligible, implying that our prior closely approximates a Gaussian distribution. **(ii):** we propose the partial sampling strategy in our framework, which perturbs the latent at a middle timestep. It can save much time without compromising quality. **(iii):** Extensive experiments validate the effectiveness of **FreqPrior**. Specifically, our approach improves both video quality and semantic quality, achieving the highest total score over baselines on VBench (Huang et al., 2024b).

## 2 RELATED WORK

**Video generative models**   In the field of video generation, previous work has explored a range of methods, including VAEs (Kingma & Welling, 2014; Hsieh et al., 2018; Bhagat et al., 2020), GANs (Goodfellow et al., 2014; Tian et al., 2021; Brooks et al., 2022; Skorokhodov et al., 2022), and auto-regressive models (Wu et al., 2021; 2022a; Ge et al., 2022; Hong et al., 2023). Recently, diffusion models (Ho et al., 2020; Song et al., 2021b; Sohl-Dickstein et al., 2015; Dhariwal & Nichol, 2021) have showcased great abilities in image synthesis (Rombach et al., 2022; Saharia et al., 2022; Nichol et al., 2022), and pave the way towards video generation (Ho et al., 2022b; He et al., 2022; Voleti et al., 2022). Many recent works (Ho et al., 2022a; Blattmann et al., 2023; Ge et al., 2023; Guo et al., 2024; Wang et al., 2023; Chen et al., 2023) on video synthesis are text-to-video diffusion paradigm with text as a highly intuitive and informative instruction. Both ModelScope (Wang et al., 2023; Luo et al., 2023) and VideoCrafter (Chen et al., 2023) are built upon on the UNet (Ronneberger et al., 2015) architecture. VideoCrafter adds a temporal transformer after a spatial transformer in each block, while in ModelScopoe each block comprises spatial and temporal convolution layers, along with spatial and temporal attention layers. AnimateDiff (Guo et al., 2024) generates videos by integrating Stable Diffusion (Rombach et al., 2022) with motion modules.

**Noise prior for diffusion models**   Given inherent high correlations within video data, several studies (Ge et al., 2023; Qiu et al., 2024; Chang et al., 2024; Gu et al., 2023; Mao et al., 2024; Wu et al., 2024) have delved into the realm of noise prior within diffusion models. Both FreeNoise (Qiu et al., 2024) and VidRD (Gu et al., 2023) focus on initialization strategies for long video generation, with FreeNoise employing a shuffle strategy to create noise sequences with long-range relationships, while VidRD utilizes the latent feature of the initial video clip. $\int$-noise prior interprets noise as a continuously integrated noise field rather than discrete pixel values (Chang et al., 2024). However, it focuses on low-level features, making it more suitable for tasks such as video restoration and video editing. Mao et al. (2024) identifies that some pixel blocks of initial noise correspond to certain concepts, enabling semantic-level generation. Nevertheless, collecting these blocks for different concepts is time-consuming, which limits its practical application. Motivated by correlations in the noise maps corresponding to different frames, PYoCo (Ge et al., 2023) carefully designs mixed noise prior and progressive noise prior. FreeInit (Wu et al., 2024) identifies signal leakage in the low-frequency domain and uses Fourier transform to refine the noise, making the initial exploration of frequency operations in the noise prior. However, noise is essentially different from signals, making the classic frequency filtering method unsuitable. As a result, the generated videos lack motion dynamics and imaging details due to the variance decay issue. To address these limitations, we propose a novel prior to enhance the overall quality of generated videos.

## 3 METHOD

**FreqPrior** comprises three key stages: **sampling process**, **diffusion process**, and **noise refinement**, as shown in Figure 2. To obtain a new noise prior, our method starts with Gaussian noise, which then goes through these three stages sequentially, repeated several times, to result in a refined noise prior. Once the new prior is established, it serves as the initial latent for video diffusion models to generate a video. The **sampling process** in our framework is DDIM sampling (Song et al., 2021a).

### 3.1 DIFFUSION PROCESS

During the sampling process, the latent becomes clean. Unlike the conventional diffusion process that typically diffuses the clean latent to timestep $T$, our approach perturbs the latent with the initial noise $\epsilon$ once sampling reaches a specific intermediate timestep, denoted as $t$. The diffusion process can be formulated as follows by leveraging the Markov property:

$$\mathbf{z}^i_{noise} = \sqrt{\frac{\bar{\alpha}_T}{\bar{\alpha}_t}}\mathbf{z}^i_t + \sqrt{1 - \frac{\bar{\alpha}_T}{\bar{\alpha}_t}}\epsilon, \tag{1}$$

where $\{\bar{\alpha}_j\}_{j=0}^T$ are the notations corresponding to the diffusion scheduler (Ho et al., 2020), and $i$ represents the $i$-th iteration.

The rationale for conducting the diffusion process beforehand stems from the observation that when $t$ reaches about timestep $400$, the latent $\mathbf{z}^i_t$ has roughly taken shape and resembles the clean latent

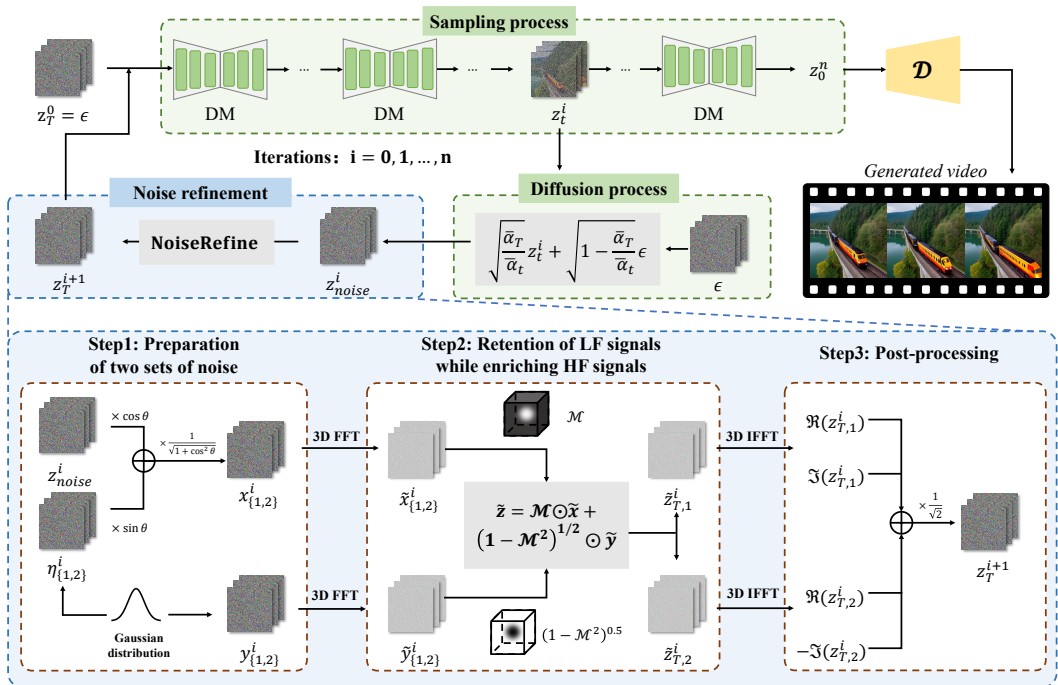

Figure 2: The framework of **FreqPrior**. It consists of three stages: **sampling process**, **diffusion process**, and **and noise refinement**. In the noise refinement stage, the noise is refined in three steps including **noise preparation**, **noise processing**, and **post-processing**.

$\mathbf{z}_0^i$, indicating the latent already has recovered large low-frequency information. Consequently, this modification yields nearly identical outcomes compared to diffusing a pure clean latent. This modification offers a notable advantage in terms of efficiency, as it significantly reduces the number of required sampling steps while maintaining consistent results. Therefore, we achieve substantial time savings without compromising the fidelity of our results.

## 3.2 NOISE REFINEMENT

The **noise refinement** stage focuses on processing different frequency components of the noise to improve video generation quality. Low-frequency signals help the model generate videos with better semantics, while high-frequency signals contribute to finer image details. Unlike conventional filtering methods, which typically target signals like images, our approach processes noise, essentially random variables, distinguishing it from traditional techniques. Therefore, we propose a novel frequency filtering method designed to effectively handle noise, enhancing overall quality.

**Step 1: Preparation of two sets of noise**   We begin by preparing two distinct sets of noise, each serving a specific purpose: one to convey low-frequency information and the other to provide high-frequency information. Initially, we independently sample from a standard Gaussian distribution to obtain $\eta_1^i$, $\eta_2^i$, $\mathbf{y}_1^i$ and $\mathbf{y}_2^i$, where $\mathbf{y}_1^i$ and $\mathbf{y}_2^i$ correspond to high-frequency information. As for low-frequency information, we combine $\mathbf{z}_{noise}^i$ with $\eta_1^i$ and $\eta_2^i$ to yield $\mathbf{x}_1^i$ and $\mathbf{x}_2^i$ as follows:

$$
\begin{aligned}
\mathbf{x}_1^i &= \frac{1}{\sqrt{1+\cos^2\theta}} \left( \cos\theta \cdot \mathbf{z}_{noise}^i + \sin\theta \cdot \eta_1^i \right), \quad \eta_1^i \sim \mathcal{N}(\mathbf{0}, \boldsymbol{I}), \\
\mathbf{x}_2^i &= \frac{1}{\sqrt{1+\cos^2\theta}} \left( \cos\theta \cdot \mathbf{z}_{noise}^i + \sin\theta \cdot \eta_2^i \right), \quad \eta_2^i \sim \mathcal{N}(\mathbf{0}, \boldsymbol{I}).
\end{aligned}
\tag{2}
$$

Here, ratio $\cos\theta$ controls the proportion of $\mathbf{z}_{noise}^i$ contained within $\mathbf{x}_1^i$ and $\mathbf{x}_2^i$. It adds flexibility to the framework, allowing us to control the amount of low-frequency information derived from $\mathbf{z}_{noise}^i$.

---

**Algorithm 1** FreqPrior

---

**Require:**

$T$: total diffusion step; $t$: middle timestep; $\{\alpha\}_{t=0}^{T}$: scheduler. $n$: number of iterations.

1: Initialize $\mathbf{z}_T = \epsilon$, where $\epsilon \sim \mathcal{N}(\mathbf{0}, \mathbf{I})$.
2: ▷ ***Obtain the noise prior***
3: **for** $i = 0$ **to** $n$ **do**
4:     $\mathbf{z}_t \leftarrow \text{Sampling}(\mathbf{z}_T)$                            ▷ Partial sampling process
5:     $\mathbf{z}_{noise} = \sqrt{\bar{\alpha}_T / \bar{\alpha}_t} \cdot \mathbf{z}_t + \sqrt{1 - \bar{\alpha}_T / \bar{\alpha}_t} \cdot \epsilon$        ▷ Diffusion Process
6:     $\mathbf{z}_T \leftarrow \text{NoiseRefine}(\mathbf{z}_{noise})$                     ▷ Noise refinement
7: ▷ ***Generate a video from new noise prior***
8: $\mathbf{z}_0 \leftarrow \text{Sampling}(\mathbf{z}_T)$                               ▷ Sampling process
9: video $\leftarrow \text{Decode}(\mathbf{z}_0)$
10: **return** video

---

**Step 2: Retention of low-frequency signals while enriching high-frequency signals** We apply the Fourier transform to map the noise to the frequency domain:

$$\tilde{\mathbf{x}}_1^i = \mathcal{F}_{3D}(\mathbf{x}_1^i), \quad \tilde{\mathbf{x}}_2^i = \mathcal{F}_{3D}(\mathbf{x}_2^i), \quad \tilde{\mathbf{y}}_1^i = \mathcal{F}_{3D}(\mathbf{y}_1^i), \quad \tilde{\mathbf{y}}_2^i = \mathcal{F}_{3D}(\mathbf{y}_2^i), \tag{3}$$

where $\mathcal{F}_{3D}$ represents the Fourier transform operation on temporal and spatial dimensions. We then perform filtering with a low-pass filter $\mathcal{M}$:

$$\tilde{\mathbf{z}}_1^i = \mathcal{M} \odot \tilde{\mathbf{x}}_1^i + (\mathbf{1} - \mathcal{M}^2)^{0.5} \odot \tilde{\mathbf{y}}_1^i, \quad \tilde{\mathbf{z}}_2^i = \mathcal{M} \odot \tilde{\mathbf{x}}_2^i + (\mathbf{1} - \mathcal{M}^2)^{0.5} \odot \tilde{\mathbf{y}}_2^i. \tag{4}$$

Since we are filtering Gaussian variables rather than real image signals, the conventional filtering approach may not be suitable. Typically, a high-pass filter is set to $(\mathbf{1} - \mathcal{M})$, we use $(\mathbf{1} - \mathcal{M}^2)^{0.5}$ instead. This adjustment is inspired by a fact in probability: if $\mathbf{u}$, $\mathbf{v} \sim \mathcal{N}(\mathbf{0}, \mathbf{I})$ are independent, then for $m \in [0, 1]$, it holds that $\mathbf{w} = m \cdot \mathbf{u} + (1 - m^2)^{0.5} \cdot \mathbf{v}$ is also standard Gaussian. In traditional filtering operations, the sum of the low-pass and high-pass filters equals one. However, in our approach, the sum of the squares of the low-pass and high-pass filters equals one. This modification enriches the high-frequency signals, maintaining the balance between low-frequency and high-frequency components. As a result, it mitigates the loss of details and motion dynamics, leading to higher fidelity in the generated videos.

**Step 3: Post-processing** After filtering, the frequency features are mapped back into the latent space, followed by post-processing to form the new noise prior $\mathbf{z}_T^{i+1}$. The process is as follows:

$$\mathbf{z}_T^{i+1} = \frac{1}{\sqrt{2}} \left( \Re\left(\mathbf{z}_{T,1}^i\right) + \Im\left(\mathbf{z}_{T,1}^i\right) + \Re\left(\mathbf{z}_{T,2}^i\right) - \Im\left(\mathbf{z}_{T,2}^i\right) \right), \quad \mathbf{z}_{T,\{1,2\}}^i = \mathcal{F}_{3D}^{-1}(\tilde{\mathbf{z}}_{\{1,2\}}^i). \tag{5}$$

Unlike traditional methods that overlook the imaginary component, our approach recognizes the importance of the information contained within these imaginary parts, which are crucial for preserving the variance in the noise prior. Consequently, we retain both the real and imaginary components. Specifically, we take both the positive real parts of $\mathbf{z}_{T,1}^i$ and $\mathbf{z}_{T,2}^i$, but for imaginary components, we take the positive imaginary part of $\mathbf{z}_{T,1}^i$ and the negative imaginary part of $\mathbf{z}_{T,2}^i$. This is the reason we prepare two sets of noise in **Step 1**. This symmetric formulation enhances the retention of valuable information while effectively eliminating unnecessary and complex terms.

In summary, our framework comprises two phases: the first phase focuses on finding a new noise prior, while the second phase generates a video based on that prior. The process of finding the noise prior includes the sampling process, diffusion process, and noise refinement, as previously discussed. Our framework is detailed in Algorithm 1.

### 3.3 ANALYSIS ON THE DISTRIBUTION OF DIFFERENT NOISE PRIOR

For the mixed noise prior proposed in PYoCo (Ge et al., 2023), it is constructed as follows:

$$\mathbf{z}_j = \frac{1}{\sqrt{2}} \epsilon_j + \frac{1}{\sqrt{2}} \epsilon_{share}, \quad \epsilon_j, \epsilon_{share} \sim \mathcal{N}(\mathbf{0}, \mathbf{I}), \tag{6}$$

where $\mathbf{z}_j$ and $\epsilon_j$ represent the $j$-th frame of latent $\mathbf{z}$ and Gaussian noise $\epsilon$. The noise prior $\mathbf{z}$ has correlations in the frame dimension, as each frame consists of shared noise $\epsilon_{shared}$:

$$\text{Cov}(\mathbf{z}_i, \mathbf{z}_j) = 0.5\boldsymbol{I}, \quad i \neq j. \tag{7}$$

Therefore, considering only the frame dimension, the diagonal elements of the covariance matrix are 1, and others are 0.5, which deviates standard Gaussian distribution. Similarly, the distribution of progressive noise prior also deviates from standard Gaussian distribution.

To conduct a theoretical analysis for FreeInit (Wu et al., 2024) and our method, we first need to determine the distribution of the refined noise. We begin with the following assumption:

**Assumption 1.** *After the diffusion process, $\mathbf{z}_{noise}$ follows a standard Gaussian distribution $\mathcal{N}(0, I)$.*

We focus on the frame, height, and width dimensions, as other dimensions do not affect analysis. The noise prior of FreeInit (Wu et al., 2024) has the following distribution (see Appendix B.1):

$$\mathbf{z} \sim \mathcal{N}\left(\mathbf{0}, \boldsymbol{P}^2 + (\boldsymbol{I} - \boldsymbol{P})^2\right), \quad \boldsymbol{P} = \frac{1}{N}\left(\boldsymbol{A}\boldsymbol{\Lambda}\boldsymbol{A} + \boldsymbol{B}\boldsymbol{\Lambda}\boldsymbol{B}\right), \tag{8}$$

where $\mathbf{z} \in \mathbb{R}^{fhw}$ is the vector form of the noise prior, $N$ is the length of $\mathbf{z}$, $\boldsymbol{\Lambda}$ is the diagonal matrix corresponding to the low-pass filter $\mathcal{M}$, and $\boldsymbol{A}$ and $\boldsymbol{B}$ represent real and imaginary parts of 3D Fourier matrix as illustrated in Appendix A.2.

Similarly, the distribution of our method is as follows (see Appendix B.2):

$$\mathbf{z} \sim \mathcal{N}\left(\mathbf{0}, \boldsymbol{I} - \frac{2\cos^2\theta}{1 + \cos^2\theta}\boldsymbol{Q}^2\right), \quad \boldsymbol{Q} = \frac{1}{N}\left(\boldsymbol{A}\boldsymbol{\Lambda}\boldsymbol{B} + \boldsymbol{B}\boldsymbol{\Lambda}\boldsymbol{A}\right). \tag{9}$$

To measure the deviations of two Gaussian distributions, we introduce the concept of covariance error.

**Definition 3.1** (Covariance error). *For two Gaussian variables with the same expectations, $\mathcal{N}(\mu, \boldsymbol{\Sigma}_1)$ and $\mathcal{N}(\mu, \boldsymbol{\Sigma}_2)$, the covariance error is defined as the Frobenius norm of the difference between their covariance matrices: $||\boldsymbol{\Sigma}_1 - \boldsymbol{\Sigma}_2||_F$.*

Under the condition of the same low-pass filter $\mathcal{M}$, we can derive the relationship of the covariance error of FreeInit and our method by using Equation 61 and Theorem C.1:

$$||\boldsymbol{I} - \boldsymbol{\Sigma}_{FreqPrior}||_F \leq \frac{\cos^2\theta}{1 + \cos^2\theta}||\boldsymbol{I} - \boldsymbol{\Sigma}_{FreeInit}||_F. \tag{10}$$

This inequality indicates that $1 - \frac{||\boldsymbol{I} - \boldsymbol{\Sigma}_{FreqPrior}||_F}{||\boldsymbol{I} - \boldsymbol{\Sigma}_{FreeInit}||_F} \geq \frac{1}{1 + \cos^2\theta} \geq 50\%$. This demonstrates that the refined noise produced by our method is closer to a standard Gaussian distribution. Our approach can theoretically reduce the covariance error by at least 50% compared to FreeInit (Wu et al., 2024). To further investigate the covariance error, we conduct numerical experiments with three different shapes and two types of low-pass filters: the Butterworth filter and the Gaussian filter. All computations are performed with float64 precision.

Table 1: **Numerical experiments on covariance error.** We report the covariance errors for three types of prior under various settings, including three different latent shapes and two different filters. The mixed noise prior is independent of filters.

| Prior | (16, 20, 20) | | (16, 30, 30) | | (16, 40, 40) | |
|---|---|---|---|---|---|---|
| | Butterworth | Gaussian | Butterworth | Gaussian | Butterworth | Gaussian |
| Mixed | 154.9193 | | 232.3790 | | 309.8387 | |
| FreeInit | 3.8230 | 8.5878 | 5.7001 | 12.8817 | 7.6026 | 17.1756 |
| Ours | $8.5071 \times 10^{-28}$ | $7.7218 \times 10^{-28}$ | $1.4002 \times 10^{-26}$ | $1.2656 \times 10^{-26}$ | $2.7342 \times 10^{-26}$ | $2.4140 \times 10^{-26}$ |

As illustrated in Table 1, our proposed noise prior exhibits the lowest covariance errors, which are minimal and can be considered negligible. FreeInit shows some covariance errors, indicating the presence of a variance decay issue. The covariance errors for the mixed noise prior are significantly higher, suggesting that it deviates substantially from a standard Gaussian distribution. These numerical experiments imply that our noise prior can be regarded as a standard Gaussian distribution.

Table 2: **Main results.** For different types of noise prior, we provide the settings for finding the prior and sampling steps for video generation. Evaluation metrics include *quality score*, *semantic score*, and *total score*. Additionally, we report the inference time, which includes the time for finding the noise prior and the time for generation.

| Base model | Noise prior | Prior finding | Generation | Quality | Semantic | Total | Inference time |
|---|---|---|---|---|---|---|---|
| VideoCrafter | Gaussian | / | 25 steps | 69.50 | 54.92 | 66.58 | **27.73s** |
| | Mixed | / | 25 steps | – | – | – | – |
| | Progressive | / | 25 steps | – | – | – | – |
| | Gaussian | / | 3*25 steps | 69.75 | 58.10 | 67.42 | 83.09s |
| | FreeInit | 2 full sampling | 25 steps | 70.62 | 58.97 | 68.29 | 83.18s |
| | Ours | 2 partial sampling | 25 steps | **70.63** | **61.33** | **68.77** | 63.67s |
| ModelScope | Gaussian | / | 50 steps | 73.13 | 65.69 | 71.64 | **19.24s** |
| | Mixed | / | 50 steps | – | – | – | – |
| | Progressive | / | 50 steps | – | – | – | – |
| | Gaussian | / | 3*50 steps | 73.25 | 66.31 | 71.87 | 57.72s |
| | FreeInit | 2 full sampling | 50 steps | 73.61 | 67.24 | 72.34 | 57.73s |
| | Ours | 2 partial sampling | 50 steps | **74.04** | **69.06** | **73.04** | 44.88s |
| AnimateDiff | Gaussian | / | 25 steps | 79.56 | 69.03 | 77.45 | **23.34s** |
| | Mixed | / | 25 steps | – | – | – | – |
| | Progressive | / | 25 steps | – | – | – | – |
| | Gaussian | / | 3*25 steps | 79.49 | 69.71 | 77.54 | 70.22s |
| | FreeInit | 2 full sampling | 25 steps | 79.58 | 68.85 | 77.43 | 70.45s |
| | Ours | 2 partial sampling | 25 steps | **80.05** | **70.37** | **78.11** | 54.05s |

## 4 EXPERIMENTS

### 4.1 EXPERIMENTAL SETTINGS

**Baselines** In our experiments, we establish the following baselines: Gaussian noise, mixed noise, progressive noise, and FreeInit (Wu et al., 2024). Gaussian noise serves as the default prior for diffusion models. The mixed noise prior and progressive noise prior are proposed by PYoCo (Ge et al., 2023). FreeInit is the pioneering work that employs Fourier transform to create a new prior.

**Implementations** We conduct the experiments on three open-soruce text-to-video diffusion models: VideoCrafter (Chen et al., 2023), ModelScope (Wang et al., 2023), and AnimateDiff (Guo et al., 2024). DDIM (Song et al., 2021a) is set to the default sampler, with the scheduler's offset configured to 1. Both FreeInit and our method require additional samplings to acquire the noise prior, with the number of extra sampling iterations set to 2. To ensure fairness, we use a Butterworth Filter with a normalized spatial-temporal cutoff frequency of 0.25 as the low-pass filter for both FreeInit and our method. In our approach, the timestep $t$ is set to 321, the ratio $\cos\theta$ is set to 0.8 for ModelScope and AnimateDiff, and 0.7 for VideoCrafter. All experiments are conducted on NVIDIA V100 GPUs. For more details, please refer to Appendix D.

**Evaluation** To evaluate the performance of each noise prior, we use VBench (Huang et al., 2024b), a comprehensive benchmark that closely aligns with human perception. VBench dissects evaluation into specific, hierarchical, and disentangled dimensions, each featuring tailored prompts and evaluation methods. Specifically, VBench assesses performance across two primary levels: *quality score* and *semantic score*. The *total score* is calculated as the weighted average of the *quality score* and *semantic score*. The scores range from 0 to 100, with a higher score indicating better performance in the corresponding aspects. For each noise prior, we generate 4730 videos for VBench evaluation. For more details, please refer to Appendix E.

### 4.2 MAIN RESULTS

**Quantitative results** As shown in Table 2, our method achieves the highest scores across all metrics, *quality score*, *semantic score*, and *total score*, on the three different base models, underscoring the superiority of our proposed noise prior. Our approach enhances both the video fidelity and se-

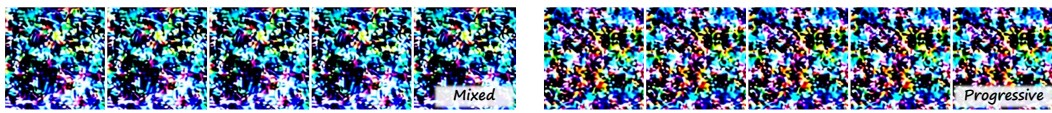

Figure 3: **Generation results using PYoCo prior.** Both mixed noise prior and progressive noise prior lead to crashes on pretrained video diffusion models.

mantic consistency of the generated videos. In contrast, the mixed noise prior and progressive noise prior lead to crashes and failure in generating normal videos, as illustrated in Figure 3. This is due to the significant gap between these types of prior and the standard Gaussian distribution, as these types of prior introduce correlations in the frame dimension. The PYoCo method requires training a model specifically on these types of prior and cannot be directly applied to pre-trained diffusion models, which limits its practical applications. FreeInit and our method require two additional samplings to acquire the noise prior, resulting in increased inference time. To investigate whether the performance improvements are attributed to more denoising steps, we triple the steps during generation for Gaussian noise. While tripling the steps for Gaussian noise provides a slight performance boost, the improvements are modest, particularly on ModelScope and AnimateDiff, where the *total score* increases by only 0.23 and 0.09, respectively. Although it shows a more significant improvement of 0.84 on VideoCrafter, its *total score* still falls well short of both FreeInit and our proposed prior. FreeInit generally enhances performance compared to Gaussian noise prior, it reduces the *semantic score* and *total score* on AnimateDiff. The reason may be the negative effects of variance decay surpass the positive effects of refinement on low-frequency signals. Our method does not have such a variance decay issue. Overall, when compared to Gaussian noise with triple steps and FreeInit, our method outperforms all metrics while requiring the least inference time, saving approximately 23%. The performance gains stem from the noise refinement stage, where we introduce a new frequency filtering method targeted at the noise. The time savings arise from diffusing the latent at an intermediate step, resulting in partial sampling that reduces several denoising steps.

**Qualitative results** Figure 4 presents a comparative visualization of the results. In the top left case, our method produces video frames with superior fidelity, featuring backgrounds reminiscent of a café, while the frames generated using Gaussian noise lack any background. FreeInit further deteriorates the result compared to Gaussian noise, blurring the area within the red box into an indistinct speck. The top right case demonstrates that the videos generated by our method exhibit finer details and better semantics. In the middle left case, our results are aesthetically superior in terms of color and brightness, while those produced by Gaussian noise appear relatively dim. The middle right case highlights that both baselines fail to generate a guitar, whereas our method successfully creates one that aligns closely with the provided text prompt. In the bottom left case, the example generated by FreeInit resembles "a cat sleeping in a bowl" rather than "a cat eating food out of a bowl." In the bottom right case, the video generated from Gaussian noise is missing a "person," while FreeInit produces an unnatural representation, lacking motion dynamics. In contrast, our method delivers the highest quality video, featuring a person walking forward. Overall, these cases illustrate that our method outperforms these types of noise prior in both quality and semantics.

Table 3: **Ablation study on the impact of ratio $\cos\theta$.** We present **total score** across various values of ratio $\cos\theta$. To eliminate the effects of timestep $t$, it is fixed to 0.

| $\cos\theta$ | VideoCrafter | ModelScope | AnimateDiff |
|---|---|---|---|
| 1.0 | 69.02 | 72.82 | 78.07 |
| 0.9 | 68.96 | 72.92 | 78.07 |
| 0.8 | 68.91 | **73.12** | **78.12** |
| 0.7 | **69.04** | 72.97 | 78.09 |

## 4.3 ABLATION STUDY

**Influence of ratio $\cos\theta$** The generation results are affected by two hyper-parameters, the ratio $\cos\theta$ in the noise refinement stage and the timestep $t$ in the diffusion process. To investigate the

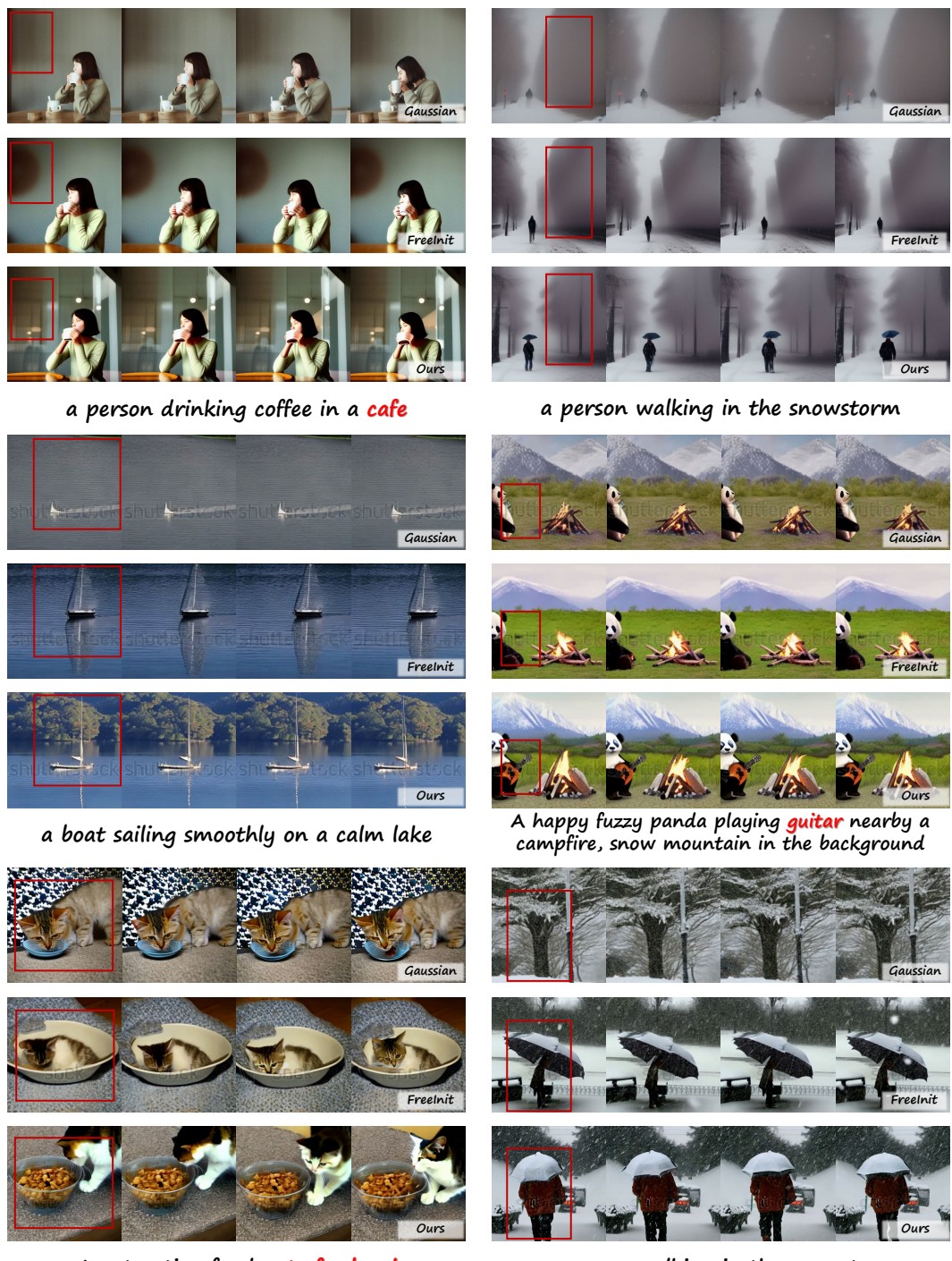

Figure 4: **Qualitative results and comparisons.** The cases in the top row are generated using AnimateDiff, while the middle row displays cases from ModelScope, and the bottom row shows cases generated by VideoCrafter. For each case, we present the generation results from different types of noise prior along with the corresponding prompt.

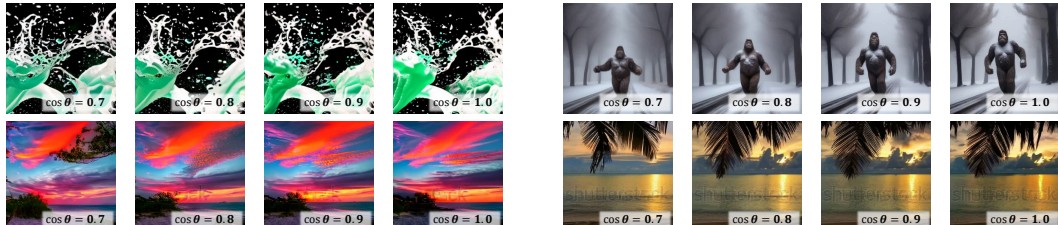

Figure 5: **Generation results on different values of $\cos\theta$.** Though there are some changes in the generated video frames as $\cos\theta$ varies, they are quite similar.

effects of $\cos\theta$, we set timestep $t$ to $0$ to eliminate the influence of $t$. In Equation (2), both $\mathbf{x}_1^i$ and $\mathbf{x}_2^i$ contribute to low-frequency signals of $\mathbf{z}_T^{i+1}$, the initial latent for the subsequent iteration. As $\cos\theta$ decreases, the proportion of $\mathbf{z}_{noise}^i$ in $\mathbf{x}_1^i$ and $\mathbf{x}_2^i$ diminishes, indicating a reduction in the low-frequency components rooted in $\mathbf{z}_{noise}^i$. $\cos\theta$ governs the extent to which low-frequency signals are retained, assuming the filter remains constant. Therefore, $\cos\theta$ can not be small. We conducted experiments with four different values of $\cos\theta$. As shown in Table 3, for AnimateDiff (Guo et al., 2024) and ModelScope (Wang et al., 2023), *Total Score* initially increases, reaching its peak at $\cos\theta = 0.8$, before declining. For VideoCrafter (Chen et al., 2023), *Total Score* get the highest at $\cos\theta = 0.7$. Overall, the differences among different $\cos\theta$ values are minor, indicating the FreqPrior is robust and not sensitive to changes in $\cos\theta$. The visualization results presented in Figure 5 demonstrate that while varying $\cos\theta$ leads to some differences in the video frames, they are still quite similar.

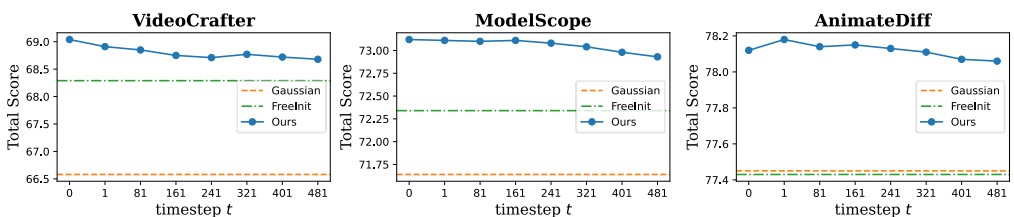

Figure 6: **Ablation study on the impact of timestep $t$.** *Total Score* is assessed accross different diffusion timesteps $t$ for three distinct text-to-video diffusion models. Overall, the timestep $t$ has little effect on the evaluated metric.

**Influence of timestep $t$.** Figure 6 shows that our method consistently outperforms Gaussian noise prior and FreeInit (Wu et al., 2024) across varying timestep $t$. While there are some fluctuations, the curve corresponding to our method on all three base models shows a slow declining trend, indicating that as the timestep $t$ increases, the quality of generated videos is likely to decrease. However, with a larger timestep, fewer denoising steps are required in each sampling iteration to find the noise prior. Consequently, it presents a trade-off between the generation quality and the inference time. Considering both factors, we selected $t = 321$ for the final setting.

## 5 CONCLUSION

We introduce a new noise prior for text-to-video diffusion models, named **FreqPrior**. The key stage in our framework lies in noise refinement, where we propose a novel frequency filtering method specifically designed for Gaussian noise. By refining the noise, we obtain a better prior for video diffusion models, thereby enhancing the quality of generation results. Although other types of noise prior have been proposed to improve the performance of video diffusion models, their distributions often deviate from a standard Gaussian distribution, leading to sub-optimal generation outcomes. In practice, the covariance error between our prior and Gaussian noise approaches zero, indicating that our noise prior closely approximates a Gaussian distribution. Extensive experiments demonstrate the superiority of our method over existing noise priors.

## ACKNOWLEDGMENT

This work was supported in part by National Natural Science Foundation of China (Grant No. 62376060).

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

## A PRELIMINARY

### A.1 DIFFUSION MODELS

**Diffusion models** (Ho et al., 2020) are a class of generative models that recover the data corrupted by the Gaussian noise through learning a reverse diffusion process. It iteratively denoises from Gaussian noise, which corresponds to learning the reverse process of a fixed Markov Chain of length $T$. The diffusion process is a Markov chain that gradually corrupts the data with Gaussian noise. For the diffusion process given the variance schedule $\beta_t$:

$$q(x_{1:T}|x_0) = \prod_{t=1}^{T} q(x_t|x_{t-1}), \qquad q(x_t|x_{t-1}) = \mathcal{N}(x_t; \sqrt{1-\beta_t}x_{t-1}, \beta_t I). \tag{11}$$

Using the Markov property, we can sample $x_t$ at an arbitrary time $t$ from $x_0$ in closed form. Let $\alpha = 1 - \beta_t$ and $\bar{\alpha}_t = \prod_{s=1}^{t} \alpha_s$, we have

$$q(x_t|x_0) = \mathcal{N}(x_t; \sqrt{\bar{\alpha}_t}x_0, (1-\bar{\alpha}_t)I). \tag{12}$$

By the Bayes' rules, $q(x_{t-1}|x_t, x_0)$ can be expressed as follows:

$$q(x_{t-1}|x_t, x_0) = \mathcal{N}(x_{t-1}; \tilde{\mu}_t(x_t, x_0), \tilde{\beta}_t I), \tag{13}$$

$$\text{where} \quad \tilde{\mu}_t(x_t, x_0) = \frac{\sqrt{\bar{\alpha}_{t-1}}\beta_t}{1-\bar{\alpha}_t}x_0 + \frac{\sqrt{\alpha_t}(1-\bar{\alpha}_{t-1})}{1-\bar{\alpha}_t}x_t \quad \text{and} \quad \tilde{\beta}_t = \frac{1-\bar{\alpha}_{t-1}}{1-\bar{\alpha}_t}\beta_t. \tag{14}$$

For the reverse process, it generates $x_0$ from $x_T$ with prior $x_T = \mathcal{N}(x_T; 0, I)$ and transitions:

$$p_\Theta(x_{t-1}|x_t) = \mathcal{N}(x_{t-1}; \mu_\Theta(x_t, t), \Sigma_\Theta(x_t, t)). \tag{15}$$

In the equation, $\Theta$ are learnable parameters of models $\epsilon_\Theta$ which are trained to minimize the variant of the variational bound $\mathbb{E}_{x,\epsilon \sim \mathcal{N}(0,I),t}\left[\|\epsilon - \epsilon_\Theta(x_t, t)\|^2\right]$.

### A.2 FOURIER TRANSFORM

**Discrete Fourier Transform (DFT)** is one of the most important discrete transforms used in digital signal processing including image processing. The discrete Fourier transform can be expressed as the **DFT** matrix, denoted as $\boldsymbol{F}$, defined as follows:

$$\boldsymbol{F} = \left(\omega_N^{(m-1)\cdot(n-1)}\right)_{N \times N} = \begin{bmatrix} \omega_N^{0\cdot0} & \omega_N^{0\cdot1} & \cdots & \omega_N^{0\cdot(N-1)} \\ \omega_N^{1\cdot0} & \omega_N^{1\cdot1} & \cdots & \omega_N^{1\cdot(N-1)} \\ \vdots & \vdots & \ddots & \vdots \\ \omega_N^{(N-1)\cdot0} & \omega_N^{(N-1)\cdot1} & \cdots & \omega_N^{(N-1)\cdot(N-1)} \end{bmatrix} \tag{16}$$

where $\omega_N = e^{-2\pi i/N}$ is a primitive $N$-th root of unity. The inverse transform, denoted as $\boldsymbol{F}^{-1}$ can be derived from $\boldsymbol{F}$ as its complex conjugate transpose, scaled by $\frac{1}{N}$: $\boldsymbol{F}^{-1} = \frac{1}{N}\boldsymbol{F}^*$.

The **DFT** matrix $\boldsymbol{F}$ can be decomposed into its real and imaginary parts, represented respectively by matrices $\boldsymbol{A}$ and $\boldsymbol{B}$:

$$\boldsymbol{F} = \boldsymbol{A} + \boldsymbol{B}i, \quad \boldsymbol{A} = \Re(\boldsymbol{F}), \quad \boldsymbol{B} = \Im(\boldsymbol{F}). \tag{17}$$

This decomposition simplifies the understanding of the structure and properties of the DFT matrix, providing deeper insights. Using Euler's formula, $\boldsymbol{A}$ and $\boldsymbol{B}$ can be explicitly expressed as:

$$\boldsymbol{A} = \left(\cos\left((m-1)(n-1)\theta\right)\right)_{N \times N}, \quad \boldsymbol{B} = \left(\sin\left((m-1)(n-1)\theta\right)\right)_{N \times N}, \tag{18}$$

where $\theta = -\frac{2\pi}{N}$. Notably, both $\boldsymbol{A}$ and $\boldsymbol{B}$ are real symmetric matrices.

**Lemma A.1.** *For $\theta = -\frac{2\pi}{N}$ where $N$ is a positive integer, it holds that $\sum_{k=1}^{N} \sin\left(l\left(k-1\right)\theta\right) = 0$ for any integer $l$.*

*Proof.* By applying Euler's formula, we rewrite $\sin\left(l\left(k-1\right)\theta\right)$ as $\Im\left(\omega_N^{l(k-1)}\right)$, where $\omega_N = e^{-2\pi i/N}$. Then we have:

$$\sum_{k=1}^{N} \sin\left(l\left(k-1\right)\theta\right) = \sum_{k=0}^{N-1} \Im\left(\omega_N^{lk}\right) = \Im\left(\sum_{k=0}^{N-1} \omega_N^{lk}\right). \tag{19}$$

The term $\sum_{k=0}^{N-1} \omega_N^{lk}$ is the sum of geometric sequence. If $\omega_N^l = 1$, then $\sum_{k=0}^{N-1} \omega_N^{lk} = N$, yielding $\Im\left(\sum_{k=0}^{N-1} \omega_N^{lk}\right) = 0$.

Otherwise, if $\omega_N^l \neq 1$, we have $\sum_{k=0}^{N-1} \omega_N^{lk} = \left(1 - \omega_N^{lN}\right) / \left(1 - \omega_N^l\right)$. Since $\omega_N^N = 1$, then $\sum_{k=0}^{N-1} \omega_N^{lk} = 0$, and consequently $\Im\left(\sum_{k=0}^{N-1} \omega_N^{lk}\right) = 0$.

In conclusion, we have shown that $\sum_{k=1}^{N} \sin\left(l\left(k-1\right)\theta\right) = 0$ for any integer $l$. $\qquad\square$

This lemma offers foundational insights into the behavior of the sum of sinusoidal functions, Now, we introduce a theorem regarding properties of the **DFT** matrix.

**Theorem A.2.** *Given a **DFT** matrix $\boldsymbol{F} \in \mathbb{C}^{N \times N}$, with $\boldsymbol{A}$ and $\boldsymbol{B}$ representing its real and imaginary parts respectively, it holds that $\boldsymbol{A}\boldsymbol{B} = \boldsymbol{B}\boldsymbol{A} = \boldsymbol{0}$ and $\boldsymbol{A}^2 + \boldsymbol{B}^2 = N\boldsymbol{I}$.*

*Proof.* Using the property of the inverse Fourier transform, we have

$$\boldsymbol{I} = \boldsymbol{F}\boldsymbol{F}^{-1} = \frac{1}{N}\boldsymbol{F}\boldsymbol{F}^* = \frac{1}{N}\left(\boldsymbol{A} + \boldsymbol{B}i\right)\left(\boldsymbol{A} - \boldsymbol{B}i\right) = \frac{1}{N}\left(\boldsymbol{A}^2 + \boldsymbol{B}^2 - \boldsymbol{A}\boldsymbol{B}i + \boldsymbol{B}\boldsymbol{A}i\right). \tag{20}$$

Comparing real parts and imaginary parts of both sides, we derive:

$$\boldsymbol{A}^2 + \boldsymbol{B}^2 = N\boldsymbol{I}, \quad \boldsymbol{B}\boldsymbol{A} = \boldsymbol{A}\boldsymbol{B}. \tag{21}$$

Considering the matrix $\boldsymbol{A}\boldsymbol{B}$, we calculate the value of the element in the $m$-th row and $n$-th column:

$$
\begin{aligned}
\left(\boldsymbol{A}\boldsymbol{B}\right)_{mn} &= \sum_{k=1}^{N} \cos\left(\left(m-1\right)\left(k-1\right)\theta\right)\sin\left(\left(k-1\right)\left(n-1\right)\theta\right) \\
&= \frac{1}{2}\sum_{k=1}^{N}\left(\sin\left(\left(m+n-2\right)(k-1)\theta\right) - \sin\left(\left(m-n\right)(k-1)\theta\right)\right) = 0.
\end{aligned} \tag{22}
$$

The last equation holds using Lemma A.1. The equation holds for each element of $\boldsymbol{A}\boldsymbol{B}$. Therefore $\boldsymbol{A}\boldsymbol{B} = \boldsymbol{B}\boldsymbol{A} = \boldsymbol{0}$. $\qquad\square$

For the 3D Fourier transform, it can be represented as follows using the Kronecker product:

$$\boldsymbol{F}_{3D} = \boldsymbol{F}_T \otimes \boldsymbol{F}_H \otimes \boldsymbol{F}_W. \tag{23}$$

The inverse transform is given by:

$$\boldsymbol{F}_{3D}^{-1} = \frac{1}{N_T N_H N_W}\boldsymbol{F}_{3D}^*. \tag{24}$$

Similarly, we decompose $\boldsymbol{F}_{3D}$ into its real part $\boldsymbol{A}_{3D}$ and imaginary part $\boldsymbol{B}_{3D}$:

$$
\begin{aligned}
\boldsymbol{F}_{3D} &= \left(\boldsymbol{A}_T + \boldsymbol{B}_T i\right) \otimes \left(\boldsymbol{A}_H + \boldsymbol{B}_H i\right) \otimes \left(\boldsymbol{A}_W + \boldsymbol{B}_W i\right), \\
\boldsymbol{A}_{3D} &= \boldsymbol{A}_T \otimes \boldsymbol{A}_H \otimes \boldsymbol{A}_W - \boldsymbol{A}_T \otimes \boldsymbol{B}_H \otimes \boldsymbol{B}_W - \boldsymbol{B}_T \otimes \boldsymbol{A}_H \otimes \boldsymbol{B}_W - \boldsymbol{B}_T \otimes \boldsymbol{B}_H \otimes \boldsymbol{A}_W, \\
\boldsymbol{B}_{3D} &= \boldsymbol{A}_T \otimes \boldsymbol{A}_H \otimes \boldsymbol{B}_W + \boldsymbol{A}_T \otimes \boldsymbol{B}_H \otimes \boldsymbol{A}_W + \boldsymbol{B}_T \otimes \boldsymbol{A}_H \otimes \boldsymbol{A}_W - \boldsymbol{B}_T \otimes \boldsymbol{B}_H \otimes \boldsymbol{B}_W.
\end{aligned} \tag{25}
$$

By Theorem A.2 and the property or Kronecker product, it still holds that:

$$\boldsymbol{A}_{3D}^2 + \boldsymbol{B}_{3D}^2 = N_T N_H N_W \boldsymbol{I}, \quad \boldsymbol{B}_{3D}\boldsymbol{A}_{3D} = \boldsymbol{A}_{3D}\boldsymbol{B}_{3D} = \boldsymbol{0}. \tag{26}$$

It reveals that the 3D DFT matrix shares the same properties as the ordinary DFT matrix. For convenience, we denote the DFT matrix, including multi-dimensional cases as $\boldsymbol{F}$, with size denoted as $N$. Employing mathematical induction, we can extend Theorem A.2 from one-dimensional case to arbitrary finite dimensions:

**Theorem A.3.** *Given a **DFT** matrix or multi-dimension **DFT** matrix $\boldsymbol{F} \in \mathbb{C}^{N \times N}$, with $\boldsymbol{A}$ and $\boldsymbol{B}$ are its real part and imaginary part respectively, it holds that $\boldsymbol{A}\boldsymbol{B} = \boldsymbol{B}\boldsymbol{A} = \boldsymbol{0}$ and $\boldsymbol{A}^2 + \boldsymbol{B}^2 = N\boldsymbol{I}$.*

## B NOISE DISTRIBUTION ANALYSIS

### B.1 FREEINIT

FreeInit (Wu et al., 2024) uses conventional frequency filtering methods to manipulate noise, which is the key step in the framework. This step can be formulated as follows:

$$\mathbf{z}_T = \Re\left(\mathcal{F}_{3D}^{-1}\left(\mathcal{F}_{3D}\left(\mathbf{z}_{noise}\right) \odot \mathcal{M} + \mathcal{F}_{3D}\left(\eta\right) \odot \left(\mathbf{1} - \mathcal{M}\right)\right)\right), \tag{27}$$

where $\mathcal{F}_{3D}$ is the Fourier transform applied to both spatial and temporal dimensions. $\mathcal{M}$ is a spatial-temporal low-pass filter. $\mathbf{z}_{noise}$ is noisy latent derived from corrupting the clean latent with initial Gaussian noise to timestep $T$, while $\eta$ is another Gaussian noise. For analysis, we focus solely on the spatial and temporal dimensions, ignoring the batchsize and channel dimensions. Additionally, we flatten the latent $\mathbf{z}_T \in \mathbb{R}^{f \times h \times w}$ into a vector $\mathbf{z} \in \mathbb{R}^{fhw}$. The equation 27 can be expressed in matrix form as follows:

$$\mathbf{z} = \Re\left(\boldsymbol{F}^{-1}\boldsymbol{\Lambda}_x\boldsymbol{F}\mathbf{x} + \boldsymbol{F}^{-1}\boldsymbol{\Lambda}_y\boldsymbol{F}\mathbf{y}\right), \tag{28}$$

where $\boldsymbol{F}$ is the DFT matrix of transform $\mathcal{F}_{3D}$, $\mathbf{x}$ and $\mathbf{y}$ are random vectors corresponding to $\mathbf{z}_{noise}$ and $\eta$, and $\boldsymbol{\Lambda}_x$ and $\boldsymbol{\Lambda}_y$ are diagonal matrices associated with low-pass filter $\mathcal{M}$ and high-pass filter $\mathbf{1} - \mathcal{M}$. Therefore it holds that $\boldsymbol{\Lambda}_x + \boldsymbol{\Lambda}_y = \boldsymbol{I}$. This equation can be simplified to the following form using equation 17:

$$\mathbf{z} = \frac{1}{N}\left(\boldsymbol{A}\boldsymbol{\Lambda}_x\boldsymbol{A} + \boldsymbol{B}\boldsymbol{\Lambda}_x\boldsymbol{B}\right)\mathbf{x} + \frac{1}{N}\left(\boldsymbol{A}\boldsymbol{\Lambda}_y\boldsymbol{A} + \boldsymbol{B}\boldsymbol{\Lambda}_y\boldsymbol{B}\right)\mathbf{y}, \tag{29}$$

Under the Assumption 1, $\mathbf{x} = \text{vec}(\mathbf{z}_{noise}) \sim \mathcal{N}\left(\mathbf{0}, \boldsymbol{I}\right)$, where $\mathbf{x}$ and $\mathbf{y}$ are independent. Since $\mathbf{z}$ is a linear combination of independent Gaussian random vectors, it follows that $\mathbf{z}$ is also Gaussian. To derive the distribution of $\mathbf{z}$, we only need to compute its expectation and covariance. The expectation is straightforward and given by $\mathbb{E}[\mathbf{z}] = \mathbf{0}$. The covariance of $\mathbf{z}$ can be calculated as follows:

$$\text{Cov}\left(\mathbf{z}\right) = \frac{1}{N^2}\left(\boldsymbol{A}\boldsymbol{\Lambda}_x\boldsymbol{A} + \boldsymbol{B}\boldsymbol{\Lambda}_x\boldsymbol{B}\right)^2 + \frac{1}{N^2}\left(\boldsymbol{A}\boldsymbol{\Lambda}_y\boldsymbol{A} + \boldsymbol{B}\boldsymbol{\Lambda}_y\boldsymbol{B}\right)^2. \tag{30}$$

To simplify the expression, we denote $\boldsymbol{P} = \frac{1}{N}\left(\boldsymbol{A}\boldsymbol{\Lambda}_x\boldsymbol{A} + \boldsymbol{B}\boldsymbol{\Lambda}_x\boldsymbol{B}\right)$. Then the term $\boldsymbol{A}\boldsymbol{\Lambda}_y\boldsymbol{A} + \boldsymbol{B}\boldsymbol{\Lambda}_y\boldsymbol{B}$ can be expressed using $\boldsymbol{P}$:

$$\begin{aligned}\boldsymbol{A}\boldsymbol{\Lambda}_y\boldsymbol{A} + \boldsymbol{B}\boldsymbol{\Lambda}_y\boldsymbol{B} &= \boldsymbol{A}(\boldsymbol{I} - \boldsymbol{\Lambda}_x)\boldsymbol{A} + \boldsymbol{B}(\boldsymbol{I} - \boldsymbol{\Lambda}_x)\boldsymbol{B} \\ &= \boldsymbol{A}^2 + \boldsymbol{B}^2 - (\boldsymbol{A}\boldsymbol{\Lambda}_x\boldsymbol{A} + \boldsymbol{B}\boldsymbol{\Lambda}_x\boldsymbol{B}) = N\boldsymbol{I} - N\boldsymbol{P}.\end{aligned} \tag{31}$$

The last equation follows from $\boldsymbol{A}^2 + \boldsymbol{B}^2 = N\boldsymbol{I}$, as stated in Theorem A.3. Combining Equation (30) and Equation (31), the covariance of $\mathbf{z}$ is given by:

$$\text{Cov}\left(\mathbf{z}\right) = \boldsymbol{P}^2 + (\boldsymbol{I} - \boldsymbol{P})^2. \tag{32}$$

Consequently, we obtain the distribution of $\mathbf{z}$ as follows:

$$\mathbf{z} \sim \mathcal{N}\left(\mathbf{0}, \boldsymbol{P}^2 + (\boldsymbol{I} - \boldsymbol{P})^2\right). \tag{33}$$

Due to the property of the low-pass filter $\mathcal{M}$ where each element lies between 0 to 1, both $\boldsymbol{\Lambda}_x$ and $\boldsymbol{\Lambda}_y$ are semi-definite diagonal matrices. Consequently, we can prove that both $\boldsymbol{P}$ and $\boldsymbol{I} - \boldsymbol{P}$ are semi-positive definite matrices. The covariance structure resembles $a^2 + (1-a)^2$, which is less than 1 for $a \in (0, 1)$. This indicates a difference between the distribution of $\mathbf{z}$ and the standard Gaussian distribution. We explore this further in Appendix C.

### B.2 FREQPRIOR

The noise refinement stage of our method consists of three distinct steps, which are elaborated on in Section 3.2. To facilitate further analysis, we express these steps in matrix form. The first step, **noise preparation step**, can be represented as:

$$\mathbf{x}_1 = \frac{1}{\sqrt{1 + \cos^2\theta}}\left(\cos\theta \cdot \mathbf{x} + \sin\theta \cdot \boldsymbol{\eta_1}\right), \quad \mathbf{x}_2 = \frac{1}{\sqrt{1 + \cos^2\theta}}\left(\cos\theta \cdot \mathbf{x} + \sin\theta \cdot \eta_2\right), \tag{34}$$

where $\eta_1, \eta_2 \sim \mathcal{N}\left(\mathbf{0}, \boldsymbol{I}\right)$ and are independent. Under Assumption 1, $\mathbf{x} \sim \mathcal{N}\left(\mathbf{0}, \boldsymbol{I}\right)$. Obviously, $\mathbf{x}, \eta_1$, and $\eta_2$ are independent. Both $\mathbf{x}_1$ and $\mathbf{x}_2$ are linear combinations of independent of Gaussian

random vectors. Their expectation can be computed directly: $\mathbb{E}[\mathbf{x}_1] = \mathbb{E}[\mathbf{x}_2] = \mathbf{0}$. Next, we calculate the covariance of these variables. Specifically, the covariances are given by:

$$\text{Cov}\left(\mathbf{x}_1\right) = \text{Cov}\left(\mathbf{x}_2\right) = \frac{1}{1 + \cos^2\theta}\boldsymbol{I}, \quad \text{Cov}(\mathbf{x}_1, \mathbf{x}_2) = \text{Cov}(\mathbf{x}_2, \mathbf{x}_1) = \frac{\cos^2\theta}{1 + \cos^2\theta}\boldsymbol{I}. \quad (35)$$

This implies that $\mathbf{x}_1$ and $\mathbf{x}_2$ are correlated, as they both share a component of $\mathbf{x}$ when $\cos\theta \neq 0$.

The **noise processing** and **post-processing** steps can be expressed as follows:

$$\mathbf{z}_1 = \boldsymbol{F}^{-1}\boldsymbol{\Lambda}_x\boldsymbol{F}\mathbf{x}_1 + \boldsymbol{F}^{-1}\boldsymbol{\Lambda}_y\boldsymbol{F}\mathbf{y}_1, \quad \mathbf{z}_2 = \boldsymbol{F}^{-1}\boldsymbol{\Lambda}_x\boldsymbol{F}\mathbf{x}_2 + \boldsymbol{F}^{-1}\boldsymbol{\Lambda}_y\boldsymbol{F}\mathbf{y}_2, \quad (36)$$

$$\mathbf{z} = \frac{1}{\sqrt{2}}\left(\Re\left(\mathbf{z}_1\right) + \Im\left(\mathbf{z}_1\right) + \Re\left(\mathbf{z}_2\right) - \Im\left(\mathbf{z}_2\right)\right), \quad (37)$$

where $\mathbf{y}_1, \mathbf{y}_2 \sim \mathcal{N}\left(\mathbf{0}, \boldsymbol{I}\right)$ are independent. Regarding the filters, $\boldsymbol{\Lambda}_x$ and $\boldsymbol{\Lambda}_y$ are diagonal matrices corresponding to the low-pass filter $\mathcal{M}$ and the high-pass filter $(1 - \mathcal{M})^{0.5}$.

The refined noise $\mathbf{z}$ can be expressed in a following form using equation 17:

$$\begin{aligned}
\sqrt{2}N \cdot \mathbf{z} = \quad & \left(\boldsymbol{A}\boldsymbol{\Lambda}_x\boldsymbol{A} + \boldsymbol{B}\boldsymbol{\Lambda}_x\boldsymbol{B} + \boldsymbol{A}\boldsymbol{\Lambda}_x\boldsymbol{B} - \boldsymbol{B}\boldsymbol{\Lambda}_x\boldsymbol{A}\right)\mathbf{x}_1 \\
& + \left(\boldsymbol{A}\boldsymbol{\Lambda}_y\boldsymbol{A} + \boldsymbol{B}\boldsymbol{\Lambda}_y\boldsymbol{B} + \boldsymbol{A}\boldsymbol{\Lambda}_y\boldsymbol{B} - \boldsymbol{B}\boldsymbol{\Lambda}_y\boldsymbol{A}\right)\mathbf{y}_1 \\
& + \left(\boldsymbol{A}\boldsymbol{\Lambda}_x\boldsymbol{A} + \boldsymbol{B}\boldsymbol{\Lambda}_x\boldsymbol{B} - \boldsymbol{A}\boldsymbol{\Lambda}_x\boldsymbol{B} + \boldsymbol{B}\boldsymbol{\Lambda}_x\boldsymbol{A}\right)\mathbf{x}_2 \\
& + \left(\boldsymbol{A}\boldsymbol{\Lambda}_y\boldsymbol{A} + \boldsymbol{B}\boldsymbol{\Lambda}_y\boldsymbol{B} - \boldsymbol{A}\boldsymbol{\Lambda}_y\boldsymbol{B} + \boldsymbol{B}\boldsymbol{\Lambda}_y\boldsymbol{A}\right)\mathbf{y}_2.
\end{aligned} \quad (38)$$

From the mathematical form of this expression, it is evident that the matrices preceding these random vectors share similar structures. To simplify this equation, we introduce the following notations:

$$\begin{aligned}
\text{Let}: \quad & \boldsymbol{C}_x = \boldsymbol{A}\boldsymbol{\Lambda}_x\boldsymbol{A} + \boldsymbol{B}\boldsymbol{\Lambda}_x\boldsymbol{B}, \quad \boldsymbol{D}_x = \boldsymbol{A}\boldsymbol{\Lambda}_x\boldsymbol{B} - \boldsymbol{B}\boldsymbol{\Lambda}_x\boldsymbol{A}, \\
& \boldsymbol{C}_y = \boldsymbol{A}\boldsymbol{\Lambda}_y\boldsymbol{A} + \boldsymbol{B}\boldsymbol{\Lambda}_y\boldsymbol{B}, \quad \boldsymbol{D}_y = \boldsymbol{A}\boldsymbol{\Lambda}_y\boldsymbol{B} - \boldsymbol{B}\boldsymbol{\Lambda}_y\boldsymbol{A}.
\end{aligned} \quad (39)$$

Since $\boldsymbol{A}$ and $\boldsymbol{B}$ are real symmetric matrices, and $\boldsymbol{\Lambda}_x$ and $\boldsymbol{\Lambda}_y$ are diagonal matrices, it is straightforward to prove that $\boldsymbol{C}_x$ and $\boldsymbol{C}_y$ are symmetric matrices, while $\boldsymbol{D}_x$ and $\boldsymbol{D}_y$ are skew-symmetric matrices. Using these notations, Equation (38) can be simplified as follow:

$$\sqrt{2}N \cdot \mathbf{z} = \left(\boldsymbol{C}_x + \boldsymbol{D}_x\right)\mathbf{x}_1 + \left(\boldsymbol{C}_y + \boldsymbol{D}_y\right)\mathbf{y}_1 + \left(\boldsymbol{C}_x - \boldsymbol{D}_x\right)\mathbf{x}_2 + \left(\boldsymbol{C}_y - \boldsymbol{D}_y\right)\mathbf{y}_2. \quad (40)$$

In the analysis of $\sqrt{2}N \cdot \mathbf{z}$ where $\mathbf{z}$ is a Gaussian-distributed vector, we need to calculate the expectation and covariance to determine its distribution. The expectation is given by $\mathbb{E}[\mathbf{z}] = \mathbf{0}$. The covariance can be expressed as the sum of several covariance terms related to $\mathbf{x}_1$, $\mathbf{x}_2$, $\mathbf{y}_1$ and $\mathbf{y}_2$. Specifically, the covariance of $\sqrt{2}N \cdot \mathbf{z}$ can be expressed as follows:

$$\begin{aligned}
\text{Cov}(\sqrt{2}N \cdot \mathbf{z}) = \quad & \text{Cov}\left(\left(\boldsymbol{C}_x + \boldsymbol{D}_x\right)\mathbf{x}_1\right) + \text{Cov}\left(\left(\boldsymbol{C}_x - \boldsymbol{D}_x\right)\mathbf{x}_2\right) \\
& + \text{Cov}\left(\left(\boldsymbol{C}_y + \boldsymbol{D}_y\right)\mathbf{y}_1\right) + \text{Cov}\left(\left(\boldsymbol{C}_y - \boldsymbol{D}_y\right)\mathbf{y}_2\right) \\
& + \text{Cov}\left(\left(\boldsymbol{C}_x + \boldsymbol{D}_x\right)\mathbf{x}_1, \left(\boldsymbol{C}_x - \boldsymbol{D}_x\right)\mathbf{x}_2\right) \\
& + \text{Cov}\left(\left(\boldsymbol{C}_x - \boldsymbol{D}_x\right)\mathbf{x}_2, \left(\boldsymbol{C}_x + \boldsymbol{D}_x\right)\mathbf{x}_1\right).
\end{aligned} \quad (41)$$

The covariance of $\sqrt{2}N \cdot \mathbf{z}$ consists of 6 terms, with the first four terms representing the covariance of each random vector. The last two terms are cross terms that arise due to the fact that $\mathbf{x}_1$ and $\mathbf{x}_2$ are not independent. By solving these terms, We can derive the covariance of $\mathbf{z}$.

First, we focus on the covariance terms related to $\mathbf{y}_1$ and $\mathbf{y}_2$:

$$\begin{aligned}
& \text{Cov}\left(\left(\boldsymbol{C}_y + \boldsymbol{D}_y\right)\mathbf{y}_1\right) + \text{Cov}\left(\left(\boldsymbol{C}_y - \boldsymbol{D}_y\right)\mathbf{y}_2\right) \\
& = \left(\boldsymbol{C}_y + \boldsymbol{D}_y\right)\text{Cov}\left(\mathbf{y}_1\right)\left(\boldsymbol{C}_y + \boldsymbol{D}_y\right)^\top + \left(\boldsymbol{C}_y - \boldsymbol{D}_y\right)\text{Cov}\left(\mathbf{y}_1\right)\left(\boldsymbol{C}_y - \boldsymbol{D}_y\right)^\top \quad (42) \\
& = \left(\boldsymbol{C}_y + \boldsymbol{D}_y\right)\left(\boldsymbol{C}_y - \boldsymbol{D}_y\right) + \left(\boldsymbol{C}_y - \boldsymbol{D}_y\right)\left(\boldsymbol{C}_y + \boldsymbol{D}_y\right) = 2\left(\boldsymbol{C}_y^2 - \boldsymbol{D}_y^2\right).
\end{aligned}$$

Similarly, we can infer $\text{Cov}\left(\left(\boldsymbol{C}_x + \boldsymbol{D}_x\right)\mathbf{x}_1\right) + \text{Cov}\left(\left(\boldsymbol{C}_x - \boldsymbol{D}_x\right)\mathbf{x}_2\right)$ combined with Equation (35):

$$\text{Cov}\left(\left(\boldsymbol{C}_x + \boldsymbol{D}_x\right)\mathbf{x}_1\right) + \text{Cov}\left(\left(\boldsymbol{C}_x - \boldsymbol{D}_x\right)\mathbf{x}_2\right) = \frac{2}{1 + \cos^2\theta}\left(\boldsymbol{C}_x^2 - \boldsymbol{D}_x^2\right). \quad (43)$$

Having computed the first four terms, we now turn our attention to the last two cross terms. With Equation (35), we have:

$$\text{Cov}\left((C_x + D_x)\,\mathbf{x}_1, (C_x - D_x)\,\mathbf{x}_2\right) + \text{Cov}\left((C_x - D_x)\,\mathbf{x}_2, (C_x + D_x)\,\mathbf{x}_1\right)$$
$$= (C_x + D_x)\,\text{Cov}(\mathbf{x}_1, \mathbf{x}_2)\,(C_x - D_x)^\top + (C_x - D_x)\,\text{Cov}(\mathbf{x}_2, \mathbf{x}_1)\,(C_x + D_x)^\top \tag{44}$$
$$= \frac{\cos^2\theta}{1 + \cos^2\theta}\,(C_x + D_x)^2 + \frac{\cos^2\theta}{1 + \cos^2\theta}\,(C_x - D_x)^2 = \frac{2\cos^2\theta}{1 + \cos^2\theta}\,\left(C_x^2 + D_x^2\right).$$

Substituting the expression of the covariance related to $\mathbf{x}_1$, $\mathbf{x}_2$, $\mathbf{y}_1$ and $\mathbf{y}_2$ with Equations (42, 43, 44), we can express the covariance of $\sqrt{2}N \cdot \mathbf{z}$ in the following form:

$$\text{Cov}\left(\sqrt{2}N \cdot \mathbf{z}\right) = 2\left(C_x^2 - D_x^2 + C_y^2 - D_y^2\right) + \frac{4\cos^2\theta}{1 + \cos^2\theta}D_x^2. \tag{45}$$

To further simplify this equation, we need to explore the properties of $C_x$, $C_y$, $D_x$ and $D_y$. From Theorem A.3, which establish $AB = BA = 0$ and $A^2 + B^2 = NI$. We can compute the squares of matrices $C_x$ and $D_x$ as follows:

$$C_x^2 = A\Lambda_x A^2 \Lambda_x A + B\Lambda_x B^2 \Lambda_x B,$$
$$-D_x^2 = A\Lambda_x B^2 \Lambda_x A + B\Lambda_x A^2 \Lambda_x B. \tag{46}$$

Notice that the squares of $C_x$ and $D_x$ share a similar form, differing only in the middle matrix: one is $A^2$ and the other is $B^2$. This observation inspires us to calculate $C_x^2 - D_x^2$, especially since we have established $A^2 + B^2 = NI$. Therefore, we can express it as follows:

$$C_x^2 - D_x^2 = A\Lambda_x \left(A^2 + B^2\right)\Lambda_x A + B\Lambda_x \left(B^2 + A^2\right)\Lambda_x B$$
$$= NA\Lambda_x^2 A + NB\Lambda_x^2 B, \tag{47}$$

Since $C_y$ and $D_y$ follow the same pattern with only the subscript replaced, it also holds that:

$$C_y^2 - D_y^2 = NA\Lambda_y^2 A + NB\Lambda_y^2 B. \tag{48}$$

Make use of $\Lambda_y = \left(I - \Lambda_x^2\right)^{\frac{1}{2}}$, we can conclude:

$$C_x^2 - D_x^2 + C_y^2 - D_y^2 = NA\left(\Lambda_x^2 + \Lambda_y^2\right)A + NB\left(\Lambda_x^2 + \Lambda_y^2\right)B = NA^2 + NB^2 = N^2 I. \tag{49}$$

Substituting with Equations (46) and (49), we can simplifies Equation (45) to express the covariance of $\sqrt{2}N \cdot \mathbf{z}$ as follows:

$$\text{Cov}\left(\sqrt{2}N \cdot \mathbf{z}\right) = 2N^2 I - \frac{4\cos^2\theta}{1 + \cos^2\theta}\left(A\Lambda_x B^2 \Lambda_x A + B\Lambda_x A^2 \Lambda_x B\right), \tag{50}$$

Inspired by the form of $A\Lambda_x B^2 \Lambda_x A$ and $B\Lambda_x A^2 \Lambda_x B$ which are the matrix multiplication of $A\Lambda_x B$ and $B\Lambda_x A$. We creatively construct a new matrix $Q = \frac{1}{N}\left(A\Lambda_x B + B\Lambda_x A\right)$. It is easy to prove $Q$ is a symmetric matrix. The square of $Q$ is as follows:

$$Q^2 = \frac{1}{N^2}\left(A\Lambda_x B^2 \Lambda_x A + B\Lambda_x A^2 \Lambda_x B\right). \tag{51}$$

By combining Equation (50) and Equation (51) and eliminating the constant $\sqrt{2}N$ from both sides of the equation, we can calculate the covariance of $\mathbf{z}$:

$$\text{Cov}\left(\mathbf{z}\right) = I - \frac{2\cos^2\theta}{1 + \cos^2\theta}Q^2. \tag{52}$$

Finally, we derive the distribution of $\mathbf{z}$ as follows:

$$\mathbf{z} \sim \mathcal{N}\left(\mathbf{0}, I - \frac{2\cos^2\theta}{1 + \cos^2\theta}Q^2\right). \tag{53}$$

It is clear that the covariance of our refined noise is "smaller" than $I$. However, as $AB = BA = 0$ and the diagonal elements of $\Lambda_x$ ranges from 0 to 1, it gives the intuition that $Q$ is close to $\mathbf{0}$. We make further analysis in Appendix C.

## C  COVARIANCE ERROR ANALYSIS

**Theorem C.1.** *Given two semi-positive definite matrices $C$ and $D$ satisfying $C \succeq D \succeq 0$, then $||C||_F \geq ||D||_F$ where $|| \cdot ||_F$ is Frobenius Norm.*

*Proof.* Since $||C - D||_F^2 \geq 0$, then expanding it yields:

$$||C||_F^2 + ||D||_F^2 \geq \operatorname{tr}\left(C^\top D + D^\top C\right) = 2\operatorname{tr}\left(CD\right). \tag{54}$$

The last equation holds because the $C$ and $D$ are symmetric matrices and $\operatorname{tr}(\cdot)$ is invariant under circular shifts. Then we can conclude:

$$||C||_F^2 - ||D||_F^2 \geq 2\operatorname{tr}\left(CD\right) - 2||D||_F^2 = 2\operatorname{tr}\left((C - D)\,D\right). \tag{55}$$

Using Cholesky decomposition, for semi-definite matrix $\mathbf{D}$, there exists matrix $L$ such that $D = LL^\top$. Then we can derive:

$$\operatorname{tr}\left((C - D)\,D\right) = \operatorname{tr}\left((C - D)\,LL^\top\right) = \operatorname{tr}\left(L^\top\left(C - D\right)L\right). \tag{56}$$

From the given condition $C \succeq D$, thus $C - D \succeq 0$, thus $L^\top\left(C - D\right)L$ is semi-positive. Therefore the trace of this matrix will be non-negative. Therefore $||C||_F \geq ||D||_F$.  $\square$

From Equations (33) and (53), we know the covariance of refined noise for each method:

$$\Sigma_{FreeInit} = P^2 + (I - P)^2, \quad \Sigma_{FreqPrior} = I - \frac{2\cos^2\theta}{1 + \cos^2\theta}Q^2. \tag{57}$$

Consider the same settings, including that the low-pass filters are identical, meaning $\Lambda_x$ is fixed. Since the filter $\Lambda_x$ is diagonal with its diagonal elements in the range $[0, 1]$, we have $0 \preceq \Lambda_x \preceq I$. Consequently, we obtain the following inequality for matrix $P$:

$$0 \preceq P = \frac{1}{N}\left(A\Lambda_x A + B\Lambda_x B\right) \preceq \frac{1}{N}\left(A^2 + B^2\right) = I. \tag{58}$$

Now consider the difference between $\Sigma_{FreeInit}$ and $I$:

$$I - \Sigma_{FreeInit} = 2\left(P - P^2\right) \succeq 0. \tag{59}$$

This inequality holds because $0 \preceq P \preceq I$, which implies $P^2 \preceq P$. This demonstrates that $\Sigma_{FreeInit}$ is indeed "smaller" than $I$. To conduct a further analysis of $\Sigma_{FreeInit}$ and $\Sigma_{freqinit}$, we first establish the relationship between $P$ and $Q$. Noticing that $P$ and $Q$ have similar forms, we can derive the following results by leveraging these specific forms:

$$\begin{aligned}
P^2 + Q^2 &= \frac{1}{N^2}\left(\left(A\Lambda_x A^2 \Lambda_x A + B\Lambda_x B^2 \Lambda_x B\right) + \left(A\Lambda_x B^2 \Lambda_x A + B\Lambda_x A^2 \Lambda_x B\right)\right) \\
&= \frac{1}{N}\left(A\Lambda_x^2 A + B\Lambda_x^2 B\right) \preceq \frac{1}{N}\left(A\Lambda_x A + B\Lambda_x B\right) = P.
\end{aligned} \tag{60}$$

Combining Equations (59) and (60), we obtain:

$$I - \Sigma_{FreqPrior} = \frac{2\cos^2\theta}{1 + \cos^2\theta}Q^2 \preceq \frac{2\cos^2\theta}{1 + \cos^2\theta}\left(P - P^2\right) = \frac{\cos^2\theta}{1 + \cos^2\theta}\left(I - \Sigma_{FreeInit}\right). \tag{61}$$

Then we can analyze the covariance errors (as defined in Definition 3.1) by applying Theorem C.1:

$$||I - \Sigma_{FreqPrior}||_F \leq \frac{\cos^2\theta}{1 + \cos^2\theta}||I - \Sigma_{FreeInit}||_F. \tag{62}$$

In practice, for common continuous low-pass filters, such as Butterworth filters and Gaussian filters, the corresponding function values monotonically decrease as the frequency increases. Given that $AB = BA$ and $Q = \frac{1}{N}(A\Lambda B + B\Lambda A)$, it follows that $Q^2$ is intuitively close to a zero matrix, making the covariance error nearly zero. This is further corroborated by our numerical experiments.

## D  EXPERIMENTAL DETAILS

Three open-sourced text-to-video models are used as the base models for evaluation: They are AnimateDiff (Guo et al., 2024), ModelScope (Wang et al., 2023; Luo et al., 2023), and VideoCrafter (Chen et al., 2023).

- For AnimateDiff, we use mm-sd-v15_v2 motion module along with realisticVisionV20_v20 dreambooth LoRA [1], sampling 16 frames of at a resolution of $512 \times 512$ at 8 FPS with a guidance scale of 7.5.
- For ModelScope, we utilize the modelscope-damo-text-to-video-synthesis version, sampling 16 frames at a resolution of $256 \times 256$ at 8 FPS, with a guidance scale of 9.
- For VideoCrafter, we employ the VideoCrafter-v1 base text-to-video model, sampling 16 frames at a resolution of $320 \times 320$ at 10 FPS, with a guidance scale of 12.

## E  EVALUATION METRICS

We employ VBench (Huang et al., 2024b) for evaluation, a comprehensive benchmark designed with tailored prompts and evaluation dimensions specifically aimed at assessing video generation performance. A key feature of VBench is its incorporation of human preference annotations, ensuring alignment with human perception. VBench uses a hierarchical and disentangled scoring system, breaking the overall ***total score*** into two main components: ***quality score*** and ***semantic score***. It covers 16 evaluation dimensions, with 7 contributing to ***quality score*** and 9 contributing to ***semantic score***. Each dimension is assessed using a specially designed approach, ensuring precise and meaningful evaluation of the generated videos. The assessments involve various off-the-shelf models (Caron et al., 2021; Ruiz et al., 2023; Radford et al., 2021; Li et al., 2023b; Teed & Deng, 2020; LAION-AI, 2022; Ke et al., 2021; Wu et al., 2022b; Li et al., 2023a; Huang et al., 2023; 2024a; Wang et al., 2024), and the score for each dimension is normalized on a 0 to 100 scale, based on empirical minimum and maximum values.

- ***Quality score*** is calculated as the weighted average of seven dimensions: *subject consistency*, *background consistency*, *temporal flickering*, *motion smoothness*, *dynamic degree*, *aesthetic quality*, and *imaging quality*. The weight for *aesthetic quality* is set to 2, while the other dimensions carry a weight of 1.
- ***semantic score*** is calculated as the weighted average of nine dimensions: *object class*, *multiple objects*, *human action*, *color*, *spatial relationship*, *scene*, *appearance style*, *temporal style*, and *overall consistency*, with each dimension equally weighted 1.

After calculating ***quality score*** and ***semantic score***, ***total score*** is calculated as follows:

$$\text{Total} = \frac{w_q}{w_q + w_s}\text{Quality} + \frac{w_s}{w_q + w_s}\text{Semantic}, \tag{63}$$

where $w_q$ and $w_s$ are 4 and 1 respectively by default.

## F  VISUALIZATION RESULTS

**More qualitative results.**  Additional qualitative results are presented in Figure 8. The videos generated using our noise prior exhibit superior video quality, in terms of imaging details, aesthetic aspects, and semantic coherence.

**Visualization of the influence of timestep *t*.**  As illustrated in Figure 7, the first three rows of frames, which correspond to different timesteps $t$, are almost the same. In the fourth case, there are some differences in the representation of the grape stem, highlighted by a red box. The stem is absent at the timestep of 321. At the timestep of 321, the stem is missing. The final case demonstrates more notable differences; as the timestep $t$ increases, the ice cream appears to melt, and the changes are observable on the table. The visualizations suggest two key points: first, in most instances, the

---

[1] https://huggingface.co/ckpt/realistic-vision-v20/blob/main/realisticVisionV20_v20.safetensors

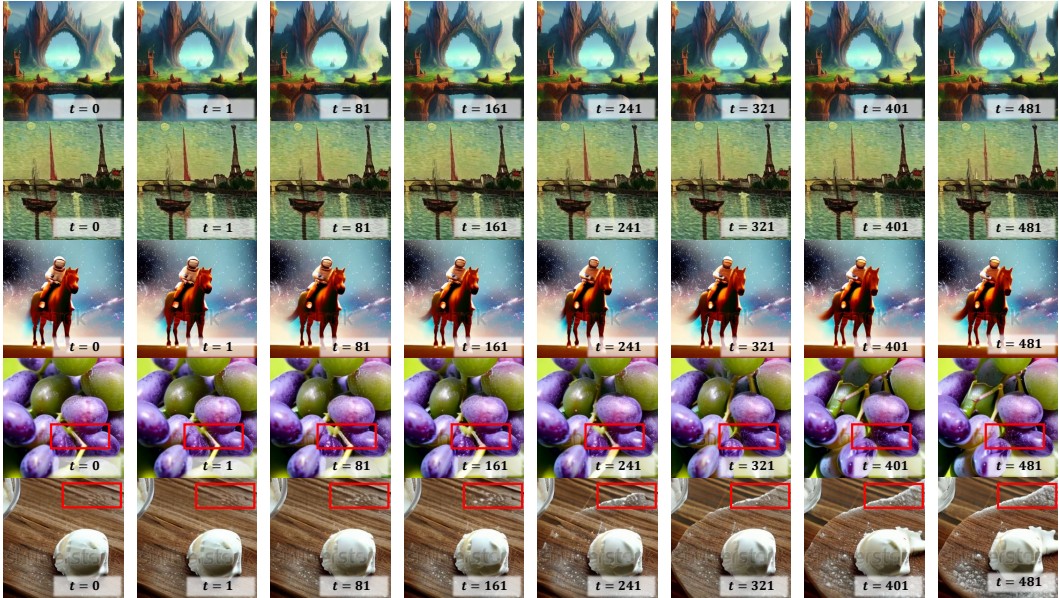

Figure 7: **Visualization results of the influence of timestep.** We present five cases where other settings are fixed to isolate the effects of varying timestep $t$. Overall, timestep $t$ has minimal impact on the generation outcomes. However, it does exert some influences on the imaging details occasionally. For the fourth and fifth cases, the red boxes highlight the differences.

timestep $t$ has minimal impact on the overall generation results; second, although the content and layout of the video frames remain largely unchanged, the timestep can indeed influence the finer imaging details. In general, the differences are quite minor, which means we can save much time by diffusing the latent at intermediate timestep during the noise refinement stage without compromising the quality of generation results.

## G    LIMITATIONS

While our method enhances consistency and smoothness in videos generated from Gaussian noise, it can occasionally result in unnatural smoothness that does not align with the laws of physics. Additionally, although our approach improves overall performance, it may alter the content layout of video frames compared to Gaussian noise. For real images, low-frequency signals typically dictate layouts; however, this is not always true for the noise prior in diffusion models. Our method refines the Gaussian noise prior using a novel frequency filtering technique, which usually preserves the structural similarity to the original Gaussian noise. Nonetheless, in some cases, the generated videos can differ significantly. During filtering, high-frequency components from other Gaussian noise may subtly change the structure of the Gaussian noise prior, resulting in variations in the content and layouts of the generated videos.

## H    BROADER IMPACTS

This work aims to propose a novel prior by refining initial Gaussian noise to enhance the quality of video generation. Text-to-video diffusion models hold the potential to revolutionize media creation and usage. While these models offer vast creative opportunities, it is crucial to address the risks of misinformation and harmful content. Before deploying these models in practice, it is essential to thoroughly investigate their design, intended applications, safety aspects, associated risks, and potential biases.

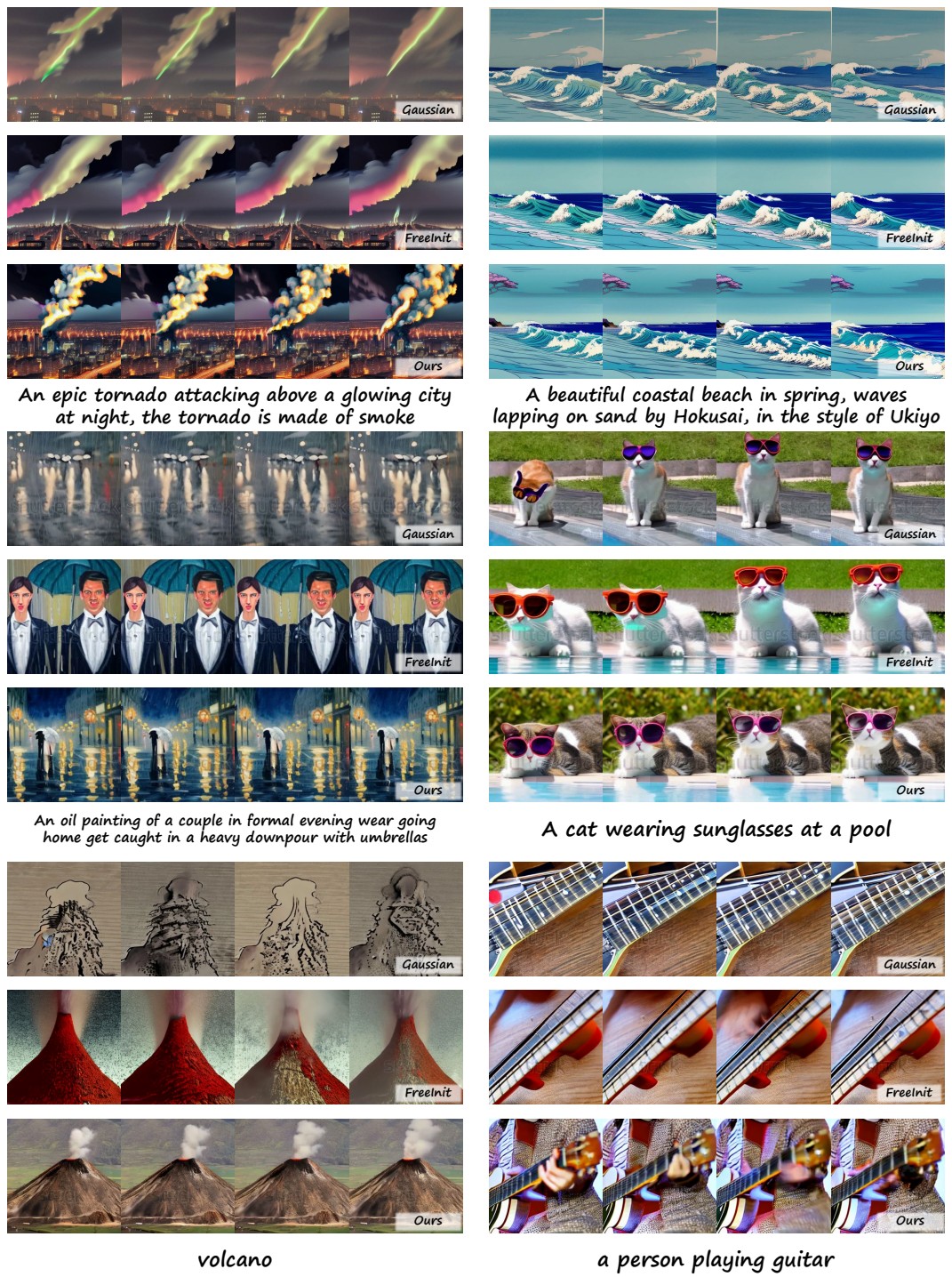

An epic tornado attacking above a glowing city at night, the tornado is made of smoke

A beautiful coastal beach in spring, waves lapping on sand by Hokusai, in the style of Ukiyo

An oil painting of a couple in formal evening wear going home get caught in a heavy downpour with umbrellas

A cat wearing sunglasses at a pool

volcano

a person playing guitar

Figure 8: **More qualitative results.**

