# OpenReview forum: "FreqPrior: Improving Video Diffusion Models with Frequency Filtering Gaussian Noise"
_ICLR.cc/2025/Conference — ICLR 2025 Poster_

### Official Review · Reviewer_PvRQ · 2024-11-02

**Soundness:** 4
**Presentation:** 4
**Contribution:** 3
**Rating:** 6
**Confidence:** 3

**Summary:**

Building on FreeInit, this method introduces a novel frequency filtering approach to obtain an improved noise prior that enhances high-frequency signals and approximates a Gaussian distribution, refining text-to-video diffusion models.
Additionally, by implementing partial sampling instead of the full sampling used in FreeInit, it effectively reduces the sampling time.

**Strengths:**

1. Comprehensive theoretical analysis of the variance decay issue of existing methods and addressing the issue by novel filtering technique is interesting and novel.

2. Extensive experiments validate the novel filtering method refine the text-to-video diffusion models significantly.

**Weaknesses:**

1. This work builds upon FreeInit, so the implementation of FreeInit should remain consistent with the original. However, while the original FreeInit uses 4 extra iterations, the comparisons in this work are made with only 2 extra iterations.

* What would the results be if both FreeInit and FreqPrior were implemented with 4 extra iterations? Would FreqPrior still outperform FreeInit?

2. Applying this method to recent T2V models could enhance the completeness of the paper.
* If high-quality T2V models are available, making low-frequency matching unnecessary, would this method still be effective?
* Additionally, if possible, could the method demonstrate effectiveness on the latest T2V models, such as T2V-Turbo or Pyramidal Flow?

**Questions:**

Questions are listed in the weakness.

---

> ### Author Response · Authors · 2024-11-22
> **Reply to Reviewer PvRQ**
>
> We thank the reviewer for the constructive feedback. We will address the remaining questions below.
>
> ---
> **Results on different iterations**
> > What would the results be if both FreeInit and FreqPrior were implemented with 4 extra iterations? Would FreqPrior still outperform FreeInit?
>
>
> We have conducted experiments in which both FreeInit and our method are implemented with 4 extra iterations. The reaults are presented in the following tables.
>
> |  Method               | Quality Score | Semantic Score | Total Score |
> |:---------------------:|:-------------:|:--------------:|:-----------:|
> | AnimateDiff + FreeInit|    77.49      |     68.35      |    77.49    |
> | AnimateDiff + Ours    |  **80.10**    |   **69.73**    |  **78.03**  |
>
> |  Method               | Quality Score | Semantic Score | Total Score |
> |:---------------------:|:-------------:|:--------------:|:-----------:|
> | ModelScope + FreeInit |    73.41	    |     67.05	     |    72.14    |
> | ModelScope + Ours     |  **74.12**	|   **69.06**	 |  **73.11**  |
>
> |  Method               | Quality Score | Semantic Score | Total Score |
> |:---------------------:|:-------------:|:--------------:|:-----------:|
> |VideoCrafter + Freeinit|    71.05	    |     58.96	     |    68.63    |
> |VideoCrafter + Ours    |  **71.16**	|   **62.45**	 |  **69.42**  |
>
> As shown in the tables, our method consistently outperforms FreeInit with the setting of 4 extra iterations, highlighting the superiority of our approach. In our paper, we opted for 2 extra iterations to balance computational time with performance improvements, as we found that this setting provides a good trade-off.
>
> ---
> **Experiments on more recent diffusion models**
> > If high-quality T2V models are available, making low-frequency matching unnecessary, would this method still be effective?
>
> We conducted experiments on OpenSora, sampling videos with 16 frames for evaluation. The results are presented in the table below:
> | Prior    | Quality Score | Semantic Score | Total Score |
> |:--------:|:-------------:|:--------------:|:-----------:|
> | Gaussian |   75.60       |    69.31       |  74.37      |
> | FreeInit |   75.98       |    69.39       |  74.66      |
> | Ours     | **75.99**     |  **69.51**     |  **74.70**  |
>
> As shown in the table, both FreeInit and our method improve performance, demonstrating the effectiveness of low-frequency matching.

---

> > ### Comment · Reviewer_PvRQ · 2024-11-23
> >
> > I sincerely appreciate the authors for taking the time to provide such a detailed response. I have some additional questions regarding the results:
> >
> > 1. Could you please provide some insights into why the performance of FreeInit decreases while the performance of the proposed method improves when the extra iterations are increased from 2 to 4?
> >
> > 2. The performance gain in the high-quality T2V model appears to be incremental compared to FreeInit. Could you explain why the performance gain seems relatively incremental for the high-quality T2V model?

---

> ### Author Response · Authors · 2024-11-23
> **Reply to Reviewer PvRQ**
>
> Thanks again for reviewer's time and efforts for reviewing our paper and providing insightful comment. Our responses to the reviewer’s concerns are below:
>
> **why the performance of FreeInit decreases while the performance of the proposed method improves?**
>
> Due to the variance decay issue in FreeInit, excessive iterations can degrade imaging details and motion dynamics, negatively affecting overall quality. While FreeInit enhances low-frequency information with each iteration, the issue of variance decay persists. With four iterations (i.e., adding two additional iterations), the negative impact of variance decay outweighs the benefits of enhanced low-frequency information, leading to a slight decrease in scores.
>
> In contrast, our method effectively addresses the variance decay issue through our novel frequency filtering approach. As shown in Table 1, the covariance error of noise prior refined by our method is less than $10^{-16}$, making it negligible. The addition of two extra iterations further enhances the low-frequency information, improving the consistency of the generated video and resulting in a increase in scores.
>
>
> **why the performance gain seems relatively incremental for the high-quality T2V model?**
>
> The incremental performance gain of our method (with high-quality T2V model OpenSora) over FreeInit could be partly ascribed to the difference of the network structure:
>
> (i) The T2V model OpenSora is based on DiT, which pathifies the latent into a sequence before passing it through the network.
> In contrast, UNet does not pathify the latent in this manner.
> This pathification can make it more challenging for the model to effectively capture different frequency information.
>
> (ii) Additionally, UNet could be more sensitive to varying frequency information.
> As FreeU [1] highlights: 'The main backbone of the U-Net primarily contributes to denoising, whereas the skip connections introduce high-frequency features into the decoder module.'
> DiT-based OpenSora, on the other hand, may be less sensitive to high-frequency information.
>
> Above two factors could explain why the performance gain in OpenSora appears to be incremental compared to FreeInit.
>
> Hope our responses clarify above thoughtful questions, and it is very much appreciated if the reviewer can kindly check our responses and provide feedback with further questions/concerns (if any). We would be more than happy to address them. Thank you!
>
> >  [1] Si, Chenyang, et al. "FreeU: Free lunch in diffusion u-net." Proceedings of the IEEE/CVF Conference on Computer Vision and Pattern Recognition. 2024.

---

> > ### Comment · Reviewer_PvRQ · 2024-11-23
> >
> > I sincerely appreciate the authors for taking the time to provide their detailed responses. All of my concerns have been thoroughly addressed. I will maintain my original rating.

---

> > > ### Author Response · Authors · 2024-11-25
> > >
> > > We appreciate the reviewer's time for reviewing and thanks again for the valuable comments and the positive score!

---

### Official Review · Reviewer_QxjB · 2024-11-04

**Soundness:** 3
**Presentation:** 3
**Contribution:** 2
**Rating:** 6
**Confidence:** 4

**Summary:**

To address the problem of variance-decreasing in FreeInit, the authors propose to re-design the low-pass filter in FreeInit and use two sets of noise to maintain the variance of intermediate diffusion variables. Experiments show that the proposed method is able to preserve more details than FreeInit.

**Strengths:**

1. The FreqPrior approach addresses detail loss and motion dynamics issues better than previous methods (e.g., FreeInit), leading to improved video fidelity.
2.  The partial sampling process significantly reduces inference time by around 23% compared to similar methods.
3.  FreqPrior achieves higher scores in quality and semantics in evaluations, especially on the VBench benchmark.

**Weaknesses:**

The paper argues that the variance-decreasing problem in FreeIniit causes it to generate over-smoothed results. But the provided evidence is weak. Although the demo cases at the beginning of this paper support this conclusion, more videos in the supplement files do not verify it.  According to Table 2,  the quantitative improvements over FreeIniit is also marginal.

**Questions:**

What is the performance on more recent ODE-based diffusion models?

---

> ### Author Response · Authors · 2024-11-22
> **Reply to Reviewer QxjB**
>
> We thank the reviewer for the constructive feedback. We will address the remaining questions below.
>
> ---
> **Improvements over FreeInit are marginal**
> >  According to Table 2, the quantitative improvements over FreeIniit is also marginal.
>
> On the **total score**, our method achieves improvements over FreeInit by 0.68, 0.70, and 0.48 on AnimateDiff, ModelScope, and VideoCrafter, respectively, **resulting in an average improvement of 0.62 on VBench—a significant gain**. The improvements of our method over FreeInit are even more pronounced in the semantic aspects. Furthermore, FreeInit achieves a total score of 77.43, which is worse than the Gaussian noise baseline (77.54), whereas our method surpasses both.
>
> Considering both the metrics and inference time (**save 23% compared to FreeInit**), our method demonstrates significant improvements over FreeInit.
>
>
> ---
> **Experiments on more recent diffusion models**
> > What is the performance on more recent ODE-based diffusion models?
>
> We conducted experiments on OpenSora, sampling videos with 16 frames for evaluation. The results are presented in the table below:
> | Prior    | Quality Score | Semantic Score | Total Score |
> |:--------:|:-------------:|:--------------:|:-----------:|
> | Gaussian |   75.60       |    69.31       |  74.37      |
> | FreeInit |   75.98       |    69.39       |  74.66      |
> | Ours     | **75.99**     |  **69.51**     |  **74.70**  |
>
> Our method achieves the highest scores across all metrics, highlighting its effectiveness.
>
> ---
> **Evidence for over smoothed results**
> > The paper argues that the variance-decreasing problem in FreeInit causes it to generate over-smoothed results. But the provided evidence is weak.
>
> In addition to the example in Figure 1, further qualitative results in Figures 4 and Figure 8 demonstrate that FreeInit tends to generate over-smoothed outputs. A representative case is shown in the top-right corner of Figure 8, where FreeInit tends to 'simplify' video frames, lacking complex image details. This issue arises due to the variance decay problem, which is caused by a lack of high-frequency information in FreeInit.
>
> Moreover, beyond the lack of imaging details, FreeInit also tends to generate videos with reduced motion dynamics. The results of motion dynamics are presented in the following table.
> |  Prior  | AnimateDiff | ModelScope | VideoCrafter |
> |:-------:|:-----------:|:----------:|:------------:|
> | Gaussian|   78.06     |   63.33    |    60.28     |
> | FreeInit|***68.06***  |***61.11*** | ***55.28***  |
> | Ours    |   75.56     |   67.22    |    62.78     |
>
> As shown in the table, FreeInit reduces motion dynamics. This supports our statement, as a lack of motion dynamics can be interpreted as over-smoothing in the temporal dimension.

---

> > ### Comment · Reviewer_QxjB · 2024-12-02
> >
> > I appreciate the authors' efforts in this work. I have read the authors' responses and other reviewers' comments. I acknowledge the contribution of this paper and I would like to improve my score from 5 to 6. However, I insist that the improvements of this paper over Freeinit are not significant enough. According to the provided comparisons on more recent models (Opensora), the improvements are further narrowed.

---

> ### Author Response · Authors · 2024-11-25
> **Reply to Reviewer QxjB**
>
> Dear Reviewer QxjB,
>
> We sincerely appreciate the reviewer's time for reviewing, and we really want to have a further discussion with the reviewer to see if our response solves the concerns. We have addressed all the thoughtful questions raised by the reviewer *(the performance of our method)* and we hope that our work's impact and results are better highlighted with our responses. It would be great if the reviewer can kindly check our responses and provide feedback with further questions/concerns (if any). We would be more than happy to address them. Thank you!
>
> Best wishes,
>
> Authors

---

### Official Review · Reviewer_PCyZ · 2024-11-04

**Soundness:** 3
**Presentation:** 3
**Contribution:** 3
**Rating:** 6
**Confidence:** 5

**Summary:**

The paper presents a novel approach for enhancing noise priors in text-to-video diffusion models. The authors introduce a new frequency filtering method to refine noise priors, improving video quality by preserving essential details and dynamics better than existing baselines such as Gaussian noise, mixed noise, progressive noise, and FreeInit. The core motivation is to keep the standard Gaussian distribution for the frequency-based sampling refinement process. The method requires additional sampling iterations but offers notable performance improvements across multiple metrics evaluated on the VBench benchmark. The experiments are conducted using three open-source text-to-video models (VideoCrafter, ModelScope, and AnimateDiff), and the results highlight that the proposed method outperforms the baselines in both quantitative and qualitative aspects.

**Strengths:**

1. This work identifies the importance of standard Gaussian distribution in the sampling process for video generation.

2. This work introduces a new frequency decomposition strategy for random variables.

3. Extensive experiments and theoretical derivation provide a great illustration for the motivation.

**Weaknesses:**

1. Despite this work has shown the side effects of non-uniformed sampling noise distribution, it is still hard to understand why this will destroy the motion dynamics from the theoretical perspectives.

2. The evaluation of this work is only based on VBench, which is somehow not sufficient. It is suggested to include more comparisons in terms of FID, FVD, etc. Whether the conclusion will stand under these metrics.

3. This work lacks user study and does not provide the detailed prompts used for video generation. Since the video quality measurement for AIGCs is not absolutely reliable, providing a user study for video generation is essential.

4. How to obtain the equation (7), it needs a detailed explanation.

**Questions:**

1. My first question is the experimental analysis, why only Vbench is provided?

2. The second question is theoretical evidence for why non-normalized Gaussian distribution will cause the worse motion dynamics.

3. Have you considered or tested other types of frequency filtering (e.g., adaptive filtering methods) to optimize the noise prior? What is the generalization capability of such frequency filtering? It would be important to demonstrate their broader applicability

4. Have you measured the standard deviation for your generated videos with different seeds? It contains lots of randomness in video generation. Whether this work select videos based on human visualization? If not, it takes which principles for results selection?

---

> ### Author Response · Authors · 2024-11-22
> **Reply to Reviewer PCyZ**
>
> We thank the reviewer for the constructive feedback. We will address the remaining questions below.
>
> ---
> **VBench**
> > My first question is the experimental analysis, why only Vbench is provided?
>
> We chose VBench for evaluation due to its distinct advantages. Specifically, there are several key reasons:
> 1. VBench divides the evaluation into two main components: **video quality** and **video-condition consistency**, offering a comprehensive and hierarchical evaluation framework.
> 2. VBench desinged compact yet representative prompts in terms of both evaluation and content categories.
> 3. VBench has conducted experiments, validating its evaluation results align with human perception.
>
>
> ---
> **Generation prompts & Standard Deviation**
> > does not provide the detailed prompts used for video generation.
> > Have you measured the standard deviation for your generated videos with different seeds? It contains lots of randomness in video generation.
>
> As mentioned above, VBench includes a carefully designed prompt suite, which comprises 946 prompts in total.
>
> For evaluation, VBench requires generating 5 different videos per prompt. As noted in our paper, we generated 4730 videos ($946\times5=4730$) for each method, with the random seed initialized to 42.
>
> Generating multiple videos per prompt helps mitigate randomness and reduce the standard deviation, resulting in more reliable evaluation results.
>
> ---
> **More comparisons**
> > It is suggested to include more comparisons in terms of FID, FVD, etc. Whether the conclusion will stand under these metrics.
>
> We have conducted more comparisons with metrics IS and FVD on UCF101 dataset.
> The results of Inception Score are shown in the following table.
> | Prior    | AnimateDiff | ModelScope | VideoCrafter |
> |:--------:|:-----------:|:----------:|:------------:|
> | Gaussian |    34.62    |    29.06   |    19.82     |
> | FreeInit |    41.54    |    33.30   |    25.54     |
> |    Ours  |  **43.01**  |  **35.51** |  **27.74**   |
>
> A higher IS value means better generation quality. Our method performs the best across these three base models.
>
> The results of Fréchet Video Distance are shown in the following table.
> | Prior    | AnimateDiff | ModelScope | VideoCrafter |
> |:--------:|:-----------:|:----------:|:------------:|
> | Gaussian | **757.96**  |   763.21   |    896.19    |
> | FreeInit |   845.86    |   693.55   |    712.62    |
> |    Ours  |   835.37    | **678.09** |  **696.01**  |
>
> A lower FVD value indicates better performance. Our method performs well on ModelScope and VideoCrafter; however, it does not enhance the generation quality on AnimateDiff, nor does FreeInit.
> FVD is calculated by comparing the distribution of generated videos to that of the ground truth videos. However, for a single prompt, the generated video can vary significantly from the ground truth, even if both are aligned with the text prompt. FreeInit and our method may alter the video content (as illustrated in the last row of Figure 4 in our paper), leading to differing values. In contrast, FVD is more suitable for evaluating image-conditioned video generation and less appropriate for text-only conditioned video generation.
>
> ---
>
> **User study**
> > Since the video quality measurement for AIGCs is not absolutely reliable, providing a user study for video generation is essential.
>
> To address this concern, we conducted a user study by randomly selecting 36 different cases generated using VBench prompts. Each base model (AnimateDiff, ModelScope, and VideoCrafter) contributed 12 cases, with each case including 3 videos generated by Gaussian noise, FreeInit, and our method. We collected feedbacks from 25 participants, who were asked to vote for each case based on two dimensions: **video quality** and **text-video alignment**.
>
>
> |        Method         | Video quality | Text-video alignment |
> |:---------------------:|:-------------:|:--------------------:|
> | AnimateDiff + Gaussian| 25.93%        | 26.67%     |
> | AnimateDiff + FreeInit| 26.54%        | 28.61%     |
> | AnimateDiff + Ours    | **47.53%**    | **44.72%** |
>
>
> |        Method         | Video quality | Text-video alignment |
> |:---------------------:|:-------------:|:--------------------:|
> |  ModelScope + Gaussian| 21.12%        | 20.41%     |
> |  ModelScope + FreeInit| 28.26%        | 27.99%     |
> |  ModelScope + Ours    | **50.62%**    | **51.60%** |
>
> |        Method         | Video quality | Text-video alignment |
> |:---------------------:|:-------------:|:--------------------:|
> |VideoCrafter + Gaussian| 15.19%        | 14.33%     |
> |VideoCrafter + FreeInit| 26.27%        | 27.16%     |
> |VideoCrafter + Ours    | **58.54%**    | **58.51%** |
>
> As shown in the table, our method outperforms both baseline method and FreeInit across three different base models in terms of both video quality and text-video alignment.

---

> > ### Author Response · Authors · 2024-11-22
> > **Reply to Reviewer PCyZ**
> >
> > **Other types of frequency filtering**
> > > Have you considered or tested other types of frequency filtering (e.g., adaptive filtering methods) to optimize the noise prior? What is the generalization capability of such frequency filtering? It would be important to demonstrate their broader applicability.
> >
> > Adaptive frequency filtering introduces additional parameters that require training, whereas our method is entirely training-free and can be seamlessly integrated into off-the-shelf video diffusion models. Moreover, we have both theoretically and empirically demonstrated the effectiveness of our approach in filtering Gaussian noise.
> >
> >
> > ---
> > **Motion dynamics**
> > >The second question is theoretical evidence for why non-normalized Gaussian distribution will cause the worse motion dynamics.
> >
> > The variance decay issue leads to a lack of details, as illustrated in Figure 1 of our paper, since reduced variance results in diminished variation. Furthermore, FreeInit also suffers from a lack of motion dynamics.
> > We speculate that the tendency of FreeInit to generate videos with insufficient motion dynamics stems from the variance decay issue. Theoretically, we derived the distribution of the FreeInit prior and confirmed that it indeed exhibits a variance decay problem.
> >
> > The results for the motion dynamics dimension in VBench are presented in the table below:
> > |  Prior  | AnimateDiff | ModelScope | VideoCrafter |
> > |:-------:|:-----------:|:----------:|:------------:|
> > | Gaussian|   78.06     |   63.33    |    60.28     |
> > | FreeInit|***68.06***  |***61.11*** | ***55.28***  |
> > | Ours    |   75.56     |   67.22    |    62.78     |
> >
> > As shown, FreeInit causes a significant loss in motion dynamics. These results support our conclusion regarding the limitations of FreeInit.
> >
> > ---
> > **Visualization results**
> > > Whether this work select videos based on human visualization? If not, it takes which principles for results selection?
> >
> > Regarding visualizations, the candidate cases are selected based on their scores on the evaluation metrics, after which we randomly sample from them.
> >
> > ---
> > **Explanation of Equation (7)**
> > > How to obtain the equation (7), it needs a detailed explanation.
> >
> >
> > As for mixed noise prior with $n$ frames, each frame of noise comprises individual noise and shared noise. For $j$-th frame, the noise prior is constructed as follows(Equation 6 in our paper):
> > $$z_j=\frac{1}{\sqrt{2}}\epsilon_j+\frac{1}{\sqrt{2}}\epsilon_{share},$$
> > where $\epsilon_1,\epsilon_2,\cdots,\epsilon_j,\cdots,\epsilon_n,\epsilon_{share}$ are independent standard Gaussian noise.
> > Therefore, the correlations come from the shared noise $\epsilon_{share}$. For $i\ne j$, we can conclude the covariance of $z_i$ and $z_j$:
> > $$\mathrm{Cov}(z_i, z_j)=\frac{1}{2}\mathrm{Cov}(\epsilon_i+\epsilon_{share}, \epsilon_j+\epsilon_{share})=\frac{1}{2}\mathrm{Cov}(\epsilon_{share}, \epsilon_{share})=0.5\mathbf{I}.$$
> > Thanks for the reviewer's advice. We will make a detailed explanation in our revised version.

---

> ### Author Response · Authors · 2024-11-25
> **Reply to Reviewer PCyZ**
>
> Dear Reviewer PCyZ,
>
> We sincerely appreciate the reviewer's time for reviewing, and we really want to have a further discussion with the reviewer to see if our response solves the concerns. We have addressed all the thoughtful questions raised by the reviewer *(user study, details of equation(7), more evaluation results, and the explanation of worse motion dynamics existed in FreeInit)* and we hope that our work's impact and results are better highlighted with our responses. It would be great if the reviewer can kindly check our responses and provide feedback with further questions/concerns (if any). We would be more than happy to address them. Thank you!
>
> Best wishes,
>
> Authors

---

> > ### Comment · Reviewer_PCyZ · 2024-11-27
> >
> > I thank the reviewer for the response from the authors. However, I am still concerned about the relationship between motion dynamics and variance delay. The author clarifies that "they first speculate that the tendency of FreeInit to generate videos with insufficient motion dynamics stems from the variance decay issue. Theoretically, they derived the distribution of the FreeInit prior and confirmed that it indeed exhibits a variance decay problem". But it is not a direct theoretical explanation for motion dynamics, right? How do we confirm that the reason for motion dynamics is from variance decay? If it can be solved better, I will improve my score.

---

> > > ### Author Response · Authors · 2024-11-28
> > > **Reply to Reviewer PCyZ**
> > >
> > > To explore the relationship between variance decay and motion dynamics, we conducted additional experiments to evaluate motion dynamics at different variance levels. Specifically, we use AnimateDiff as the base model, with the variance $\sigma^2$ ranging from $0.95^2$ to $1.00^2$. We then evaluate the motion dynamics for each level. The results are shown in the table below.
> > >
> > > | variance $\sigma^2$ | motion dynamics|
> > > |:--------:|:--------------:|
> > > | $0.95^2$ |   51.67|
> > > | $0.96^2$ |   53.06|
> > > | $0.97^2$ |   55.00|
> > > | $0.98^2$ |   63.33|
> > > | $0.99^2$ |   72.72|
> > > | $1.00^2$ |   78.06|
> > >
> > > As the variance decreases, the motion dynamics value also decreases. Since the diffusion model is typically trained on data corrupted with standard Gaussian noise, noise with lower variance introduces less variation. This reduced variation blurs the video frames and diminishes the motion dynamics. In the extreme case, if the initial noise prior is set to 0, the generated video collapses.
> > >
> > > In summary, variance decay results in reduced motion dynamics, as it causes the noise to lack the necessary variation, which is essential for preserving motion dynamics.
> > >
> > > We sincerely appreciate the reviewer for their insightful feedback and suggestions for improvement. Please feel free to let us know if anything is unclear, and we would be happy to provide further clarification.

---

> > > > ### Comment · Reviewer_PCyZ · 2024-11-28
> > > >
> > > > Thanks for the response of the authors. I will improve my score to 6. It should be any other factors affecting the motion dynamics except for variance. Expect your further investigation. Thanks.

---

> > > > > ### Author Response · Authors · 2024-11-28
> > > > > **Reply to Reviewer PCyZ**
> > > > >
> > > > > Thank the reviewer for the thoughtful feedback. We truly appreciate the reviewer's suggestion and will explore other potential factors affecting motion dynamics beyond variance. We look forward to investigating this further in our future work. Once again, thank the reviewer for the valuable insights.

---

### Official Review · Reviewer_kmCy · 2024-11-04

**Soundness:** 3
**Presentation:** 3
**Contribution:** 3
**Rating:** 6
**Confidence:** 5

**Summary:**

The paper introduces FreqPrior, a novel noise initialization strategy for text-to-video diffusion models.  FreqPrior refines noise in the frequency domain using a new filtering technique that addresses different frequency signals while maintaining a noise prior distribution close to a standard Gaussian distribution. This method helps preserve important low-frequency signals, enhancing semantic fidelity. The authors propose a partial sampling process that perturbs the latent space at an intermediate timestep during the noise prior generation. This approach significantly reduces inference time without compromising the quality of the generated videos. The paper provides a comprehensive theoretical analysis of the variance decay issue in existing methods, which contributes to the loss of details and motion dynamics. The authors show that the covariance error of their method is negligible, indicating that their noise prior closely approximates a Gaussian distribution.

**Strengths:**

The main contributions are:

FreqPrior refines noise in the frequency domain using a new filtering technique that addresses different frequency signals while maintaining a noise prior distribution close to a standard Gaussian distribution. This method helps preserve important low-frequency signals, enhancing semantic fidelity.

 The authors propose a partial sampling process that perturbs the latent space at an intermediate timestep during the noise prior generation. This approach significantly reduces inference time without compromising the quality of the generated videos.

The paper provides a comprehensive theoretical analysis of the variance decay issue in existing methods, which contributes to the loss of details and motion dynamics. The authors show that the covariance error of their method is negligible, indicating that their noise prior closely approximates a Gaussian distribution.

**Weaknesses:**

The title should explicitly mention "Video Diffusion Models" to clarify that the method is specifically designed for video generation and not applicable to image diffusion models. This will avoid any confusion and make the scope of the paper clearer to readers.

The paper should provide detailed measurements of GPU memory usage before and after applying the proposed FreqPrior method, particularly focusing on peak memory consumption. Given that 3D FFT can be memory-intensive, especially for resolutions higher than 512x512, this information is crucial for understanding the practical feasibility of the method. Include tables or graphs showing the GPU memory usage for different resolutions and compare them with the baseline methods. This will help readers assess the trade-offs between memory consumption and performance improvements.


The paper should explore the impact of different Classifier-Free Guidance (CFG) strengths when using FreqPrior. Since CFG is a common technique used in diffusion models to enhance generation quality, understanding how FreqPrior interacts with varying CFG strengths is important for practical applications.

**Questions:**

The title should explicitly mention "Video Diffusion Models" to clarify that the method is specifically designed for video generation and not applicable to image diffusion models. This will avoid any confusion and make the scope of the paper clearer to readers.

The paper should provide detailed measurements of GPU memory usage before and after applying the proposed FreqPrior method, particularly focusing on peak memory consumption. Given that 3D FFT can be memory-intensive, especially for resolutions higher than 512x512, this information is crucial for understanding the practical feasibility of the method. Include tables or graphs showing the GPU memory usage for different resolutions and compare them with the baseline methods. This will help readers assess the trade-offs between memory consumption and performance improvements.

The paper should explore the impact of different Classifier-Free Guidance (CFG) strengths when using FreqPrior. Since CFG is a common technique used in diffusion models to enhance generation quality, understanding how FreqPrior interacts with varying CFG strengths is important for practical applications.

---

> ### Author Response · Authors · 2024-11-22
> **Reply to Reviewer kmCy**
>
> We thank the reviewer for the constructive feedback. We will address the remaining questions below.
>
> ---
> **Paper title**
> > The title should explicitly mention "Video Diffusion Models" to clarify that the method is specifically designed for video generation.
>
> Thank the reviewer for the valuable suggestion. We will revise the title to explicitly include 'Video Diffusion Models', ensuring it clearly reflects the focus on video generation in our method.
>
> ---
> **GPU memory usage**
> > The paper should provide detailed measurements of GPU memory usage before and after applying the proposed FreqPrior method, particularly focusing on peak memory consumption.
>
> We have measured the peak GPU memory usage before and after applying our proposed FreqPrior. We conducted the experiments on VideoCrafter. The results of peak GPU memory consumption are provided in the following table.
>
> | video shape (f, h, w) | w/o FreqPrior | w FreqPrior | Change |
> |:---------------------:|:-------------:|:-----------:|:------:|
> |    (16, 256, 256)     |   7036.68MB   |  7039.82MB  | 3.14MB |
> |    (16, 320, 320)     |   7409.05MB   |  7413.97MB  | 4.92MB |
> |    (16, 384, 384)     |   7863.48MB   |  7870.56MB  | 7.08MB |
> |    (16, 512, 512)     |   9018.48MB   |  9031.07MB  |12.59MB |
> |    (16, 640, 640)     |  10509.17MB   | 10534.46MB  |25.29MB |
> |    (16, 768, 768)     |  12323.20MB   | 12351.51MB  |28.31MB |
> |    (16, 896, 896)     |  14469.51MB   | 14509.02MB  |39.51MB |
> |    (16, 960, 960)     |  15666.62MB   | 15710.82MB  |44.20MB |
> |  (16, 1024, 1024)     |  16945.72MB   | 16996.90MB  |51.18MB |
> |  (16, 1280, 1280)     |  22892.00MB   | 22970.64MB  |78.64MB |
>
>
> As the resolution increases, FreqPrior does lead to a slight increase in peak GPU memory usage. However, the additional memory consumption is minimal. Theoretically, the computational complexity of FFT is $O(n\log n)$, whereas that of attention is typically at least $O(n^2)$. Consequently, the majority of peak memory usage stems from the diffusion model and the inference period itself.
>
> In summary, while FreqPrior slightly increases peak memory usage, the increase is negligible ---- less than 1% of the baseline methods' peak memory usage. Therefore, we conclude that GPU memory usage is not a concern with our proposed method.
>
> ---
> **Impacts of different Classifier-Free Guidance strengths**
> > The paper should explore the impact of different Classifier-Free Guidance (CFG) strengths when using FreqPrior.
>
>
> | CFG strength  | Quality Score | Semantic Score | Total Score |
> |:-------------:|:-------------:|:--------------:|:-----------:|
> | 6.0           |  80.09        | 69.85          |    78.04    |
> | 7.5 (default) |  80.05        | 70.37          |    78.11    |
> | 9.0           |  80.08        | 70.71          |    78.20    |
> | 10.5          |  79.98        | 70.15          |    78.01    |
>
> We conducted additional experiments to evaluate the impact of Classifier-Free Guidance (CFG) strength on AnimateDiff using FreqPrior. The results indicate that the total score is not significantly sensitive to changes in CFG strength. Specifically, across the range of 6.0 to 10.5, the total score fluctuates slightly between 78.01 and 78.20.

---

> > ### Comment · Reviewer_kmCy · 2024-11-24
> > **Resolved Concerns**
> >
> > Dear Authors,
> >
> > Thank you for your detailed response. My concerns have all been addressed. Thank you.
> >
> > Best regards,

---

> > > ### Author Response · Authors · 2024-11-25
> > >
> > > We appreciate the reviewer's time for reviewing and thanks again for the valuable comments and the positive score!

---

### Meta-Review · Area_Chair_JPgY · 2024-12-21

**Metareview:**

Summary: Proposes a novel frequency filtering technique to refine noise priors, enhancing high-frequency signal preservation while maintaining a noise prior approximating a standard Gaussian distribution. The authors also introduce a partial sampling technique that reduces inference sampling times without compromising image quality.

Strengths: This is among the best papers I have reviewed as an Area Chair. It is exceptionally well-written, with a clearly defined problem statement that addresses an underexplored area: improving noise initialization as opposed to focusing on well-explored directions like architecture, training, or sampling methods. The paper presents robust theoretical derivations and experimental results, offering valuable insights into design choices. The proposed techniques deliver good results, demonstrate novelty, and contribute to reducing inference times.

Weaknesses:The paper has limited comparisons with the latest open-weight text-to-video diffusion models. While the authors provided some quantitative results in the rebuttal, I encourage them to extend their analysis by showcasing their technique on these methods and including additional qualitative results.

Reason for Acceptance:
The strengths outlined above.

**Additional Comments On Reviewer Discussion:**

The paper received 4x marginally above acceptance. All reviewer concerns have been addressed, with a singular exception of unclear relationship between motion dynamics and variance decay, raised by reviewer (PcyZ). The response and additional results are convincing, and I encourage authors to include motion dynamics and variance tables into the main paper.

---

### Decision · Program_Chairs · 2025-01-22

Accept (Poster)